# Deep Trajectory Supervision: Deep Supervision Strikes Back

**Han Wang** [1 2]  **Weijie Wang** [1 3]  **Jiaqi Liu** [1]  **Hilde Kuehne** [4 5]  **Nicu Sebe** [1]

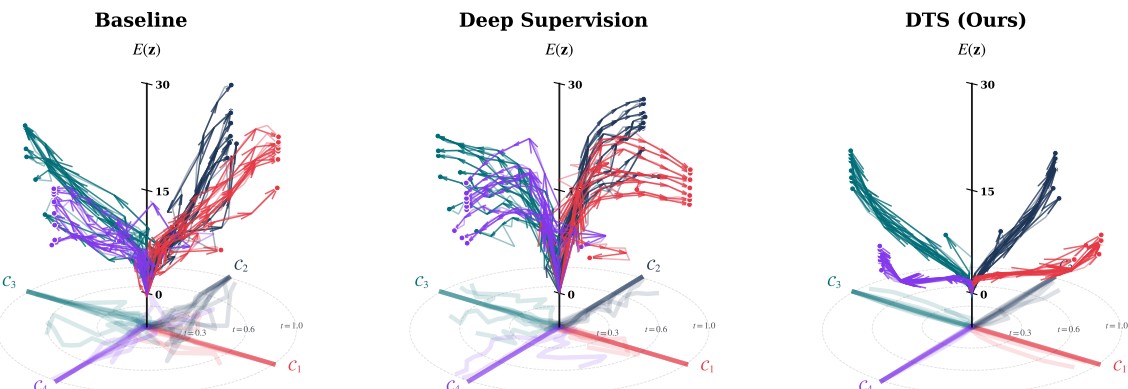

*Figure 1.* **Latent Kinematics of Conditional Discriminative Flow.** We visualize the evolution of internal representations as an input-conditioned dynamical process. These trajectories, generated from actual experimental data (ViT-Small on CIFAR-100), are projected into a cylindrical semantic space to characterize the inference flow. In this visualization, the radial distance from the origin represents the normalized depth $t \in [0, 1]$, the angular sectors denote categorical targets $\mathcal{C}_k$, and the vertical height quantifies the Semantic Evidence $E(\mathbf{z})$ as defined in Eq. (4). Distinct colors denote different classes $\mathcal{C}_k$. The bottom projections (shadows) isolate the angular evolution of the latent state, highlighting lateral drift in the representation space. **Baseline (Left):** Standard training exhibits less constrained intermediate dynamics, with irregular semantic progression during the forward pass. **Deep Supervision (Middle):** Uniform hard-label supervision can impose premature categorical confidence, causing trajectories to move toward the final target before sufficient feature maturation. **DTS (Ours, Right):** By aligning intermediate supervision with the empirically observed exponential evidence schedule, DTS encourages a smoother and more coherent semantic trajectory.

## Abstract

Conventional optimization constrains the terminal state while leaving the intermediate flow weakly regulated, overlooking the continuous-time dynamics inherent in residual architectures. In this work, we formalize the forward pass as a **Conditional Discriminative Flow** and investigate its depth-wise kinematic patterns. Using Tuned Lens analysis, we discover that the accumulation of semantic evidence follows a consistent exponential schedule, indicating that deep models naturally require an extended phase of feature construction prior to a rapid transition toward categorical certainty in the terminal layers.

Motivated by this empirical regularity, we propose **Deep Trajectory Supervision**, a framework

that aligns auxiliary supervision with this exponential evidence progression. By regulating the semantic trajectory of the inference flow, DTS provides a simple trajectory-level inductive bias for intermediate supervision. Empirical evaluations on ImageNet-1K and various benchmarks demonstrate that DTS improves convergence and terminal performance across both Transformer and CNN-style architectures, with minimal additional training overhead.

Code is released at https://github.com/Hon-Wong/DTS. [1]University of Trento, Italy [2]University of Tuebingen, Germany [3]Fondazione Bruno Kessler, Italy [4]MIT-IBM Watson AI Lab, USA [5]Tuebingen AI Center, Germany. Correspondence to: Han Wang <toameek@gmail.com>.

*Proceedings of the $43^{rd}$ International Conference on Machine Learning*, Seoul, South Korea. PMLR 306, 2026. Copyright 2026 by the author(s).

## 1. Introduction

Deep neural networks have become the de facto standard for discriminative modeling. Architectures such as Residual Networks (ResNets) (He et al., 2016), Vision Transformers (ViTs) (Dosovitskiy et al., 2021), and their variants (Xie et al., 2017; Liu et al., 2022b; 2021; Touvron et al., 2022; Yuan et al., 2021; Touvron et al., 2021) achieve this by stacking dozens of layers to build a rich feature hierarchy. The dominant training paradigm, end-to-end supervision, leverages a simple yet powerful principle: apply a strong loss signal, typically cross-entropy, exclusively to the network's terminal output. This approach effectively treats the intri-

cate, multi-stage process of feature extraction as a black box, optimizing the final input-output mapping while ignoring the intermediate transformations.

The theoretical characterization of residual architectures as discretizations of Ordinary Differential Equations (ODEs) has offered a way to open this black box (Chen et al., 2018). In this framework, the sequence of residual blocks is modeled as a discrete approximation to an underlying vector field. The forward pass for any given input can thus be viewed as a trajectory evolving through this field. The objective of classification training can therefore be re-interpreted: it is not merely to produce a correct final prediction, but also to shape the intermediate evolution that transports inputs toward semantic targets along a continuous path. However, existing training objectives primarily focus on the final state of this transport, leaving the specific path of the latent flow weakly regulated.

The significance of trajectory shape has been extensively demonstrated in generative modeling. Flow Matching (Lipman et al., 2022; Liu et al., 2022a), cast as learning an ODE velocity field, reveals that the geometric properties of probability paths are critical for optimization stability. By explicitly rectifying the transport into simpler paths, these methods simplify the learning objective. This success motivates the view that regulating the trajectory of a continuous flow can act as a useful inductive bias. However, while generative flows focus on the transport from noise to data, the kinematic patterns governing the transformation from high-entropy sensory input to categorical semantics remain less characterized. Here we want to ask:

- *Does the shape of this trajectory matter for discriminative modeling?*

- *If so, what is an appropriate path?*

To address those questions, we first formalize the discriminative process as a **Conditional Discriminative Flow (CDF)**. In this framework, the forward pass is modeled as an input-conditioned trajectory that transports initial embeddings toward a semantic target. For Vision Transformers, this is naturally represented by the layer-wise evolution of the `[CLS]` token as it interacts with the condition, *i.e.*, image $x$. For CNN-style architectures, the same perspective can be applied to the layer-wise or block-wise representations used for classification. This formalization allows us to treat internal representations as states along a continuous path, while recognizing that practical networks implement this process through discrete residual updates.

Utilizing the CDF framework, we quantify the evolution of Semantic Evidence via the Tuned Lens protocol (Belrose et al., 2023). Our analysis reveals that evidence accumulation follows a consistent exponential law, where the refinement velocity is approximately proportional to the current signal strength. This kinematic signature suggests a prolonged feature construction phase followed by a rapid transition towards certainty, providing a useful empirical basis for observations in prior classification studies (Jiang et al., 2024).

Based on these findings, we propose **Deep Trajectory Supervision (DTS)**. We attach auxiliary classification heads to the intermediate layers of the network. Instead of supervising these heads with one-hot labels (Lee et al., 2015; Jiang et al., 2024; Wang et al., 2015), DTS employs a pre-defined exponential schedule to set the target confidence at each depth. This schedule requires early layers to maintain a low semantic margin and only encourages a rapid increase in categorical certainty in the final blocks. By supervising the intermediate layers with these soft targets, DTS provides a training signal that follows the observed growth of semantic evidence. This approach avoids the problems of standard deep supervision where early layers are forced to be overly confident before they have developed necessary features.

Our experiments evaluate DTS across diverse benchmarks with different scales and semantic granularities. On ImageNet-1K (Russakovsky et al., 2015), we incorporate DTS into the competitive DeiT-III training recipe (Touvron et al., 2022). We further evaluate DTS on CNN-style architectures, including ResNets and ConvNeXts, and compare it with standard deep supervision, aligned training, and fixed-margin controls. Even within these stronger comparisons, DTS consistently improves convergence speed and final accuracy. Additional compute-normalized measurements show that the auxiliary heads introduce only minor training overhead while improving time-to-target accuracy. These results suggest that the geometry of the inference trajectory is an important factor in model performance, and that regulating the latent flow provides an effective inductive bias that complements existing optimization methods.

## 2. Related Work

**Neural ODEs.** The framework of Neural Ordinary Differential Equations (Neural ODEs) formalizes the evolution of a latent state $\mathbf{z}(t)$ as a continuous-time process governed by a parameterized vector field (E, 2017; Chen et al., 2018). Within this regime, the forward pass is interpreted as an initial value problem where the network depth represents the integration time. Each residual block serves as a discrete step of an ODE solver, such as the Euler method, mapping the input through a sequence of infinitesimal transformations (E, 2017). This perspective characterizes hidden representations as states along a continuous latent trajectory.

**Flow Matching.** In generative models, Flow Matching (Lipman et al., 2022) and Rectified Flow (Liu et al., 2022a)

show that the shape of the path is a key factor for training. These methods show that using straight lines to connect noise and data can make the model easier to train and faster to sample. This approach has been applied to more complex spaces like Riemannian manifolds and probability simplices to handle discrete categories (Chen & Lipman, 2023; Stark et al., 2024; Sriram et al., 2024; Monsefi et al., 2025). The success of generative flow matching provides the insight that explicitly regulating the trajectory shape is beneficial for convergence and performance.

**Deep Supervision.** The dominance of end-to-end supervision is largely attributed to architectural innovations such as residual connections and normalization layers, which effectively ensure gradient stability and diminish the original necessity for classical Deep Supervision to mitigate vanishing signals (Lee et al., 2015). While these designs facilitate the flow of gradients to early layers, they do not explicitly regulate the geometric properties of the latent trajectory itself. Intermediate guidance remains widely used in other domains (Wang et al., 2025; Chen et al., 2021; Zhang et al., 2022; Ren et al., 2025; Li et al., 2022). More recently, (Jiang et al., 2024) proposed aligned learning, which encourages intermediate layers to produce semantically meaningful predictions. These methods demonstrate the value of layer-wise supervision, but they typically do not specify how target confidence should evolve across depth. From the perspective of continuous latent flow, this leaves open the question of what supervision trajectory is appropriate for discriminative representation learning. Our work addresses this question by identifying an empirical progression of semantic evidence and using it to define a depth-dependent supervision schedule.

**Tuned Lens.** Probing decodes hidden representations into semantic spaces to interpret internal model dynamics (Alain & Bengio, 2016; Belrose et al., 2023; nostalgebraist, 2020). While Logit Lens utilizes fixed classification weights (nostalgebraist, 2020), Tuned Lens employs learned affine transformations to accurately estimate information content across depths (Belrose et al., 2023). Probing studies across vision (Jiang et al., 2024), language (Belrose et al., 2023), and multimodal domains (Zhang et al., 2025) reveal a consistent nonlinear progression where semantic confidence remains low before rising sharply in certain layers. This evidence motivates our approach to regulate the latent trajectory directly.

# 3. Problem Formulation: Conditional Discriminative Flow

We formalize the forward pass of deep residual architectures as an input-conditioned dynamical system. By characterizing the evolution of latent representations as a continuous approximation of discrete residual updates, we establish a framework that bridges high-dimensional latent dynamics with scalar semantic evidence. Specifically, we define the forward pass as a conditional ODE approximation, connect it with density evolution, and introduce a semantic observable to quantify evidence accumulation along the trajectory.

## 3.1. Forward Pass as a Conditional ODE

Consider a deep network with $L$ residual blocks processing an input $\boldsymbol{x} \in \mathcal{X}$. Let $\boldsymbol{z}(t) \in \mathbb{R}^d$ denote the latent state of the model at a normalized depth $t \in [0, 1]$, which represents the hidden features within the network's representation space. For Vision Transformers, $\boldsymbol{z}(t)$ specifically corresponds to the hidden state of the [CLS] token. For CNN-style architectures, it can be instantiated by the block-wise or pooled representation used for classification. We model the evolution of this latent state as a continuous approximation of first-order residual updates. In the continuous limit, the state satisfies the following initial value problem:

$$\frac{d\boldsymbol{z}(t)}{dt} = \boldsymbol{v}(\boldsymbol{z}(t), t \mid \boldsymbol{x}), \quad \boldsymbol{z}(0) = \boldsymbol{z}_{\text{init}}. \tag{1}$$

The term $\boldsymbol{z}_{\text{init}} \in \mathbb{R}^d$ denotes the initial latent state of the trajectory. We define this process as a *Conditional Discriminative Flow* (CDF). Within this framework, the condition, *i.e.*, the input image $\boldsymbol{x}$, parameterizes the velocity field $\boldsymbol{v}$. This parameterization ensures that trajectories for distinct samples evolve according to the corresponding input signal.

In practical finite-depth networks, the dynamics are implemented by discrete residual updates. Each residual block $\mathcal{F}_l(\boldsymbol{z}_l, \boldsymbol{x})$ can be viewed as a first-order Euler approximation of the local velocity with step size $\Delta t = 1/L$:

$$\boldsymbol{z}_{l+1} = \boldsymbol{z}_l + \mathcal{F}_l(\boldsymbol{z}_l, \boldsymbol{x}), \quad \mathcal{F}_l(\boldsymbol{z}_l, \boldsymbol{x}) \approx \Delta t\, \boldsymbol{v}(\boldsymbol{z}_l, t_l \mid \boldsymbol{x}). \tag{2}$$

This continuous view is therefore used as an interpretive approximation to the layer-wise residual dynamics, rather than as a claim that practical finite-depth networks exactly satisfy a continuum-limit ODE. A more explicit discrete approximation and its first-order error term are provided in Appendix A.2.

## 3.2. Semantic Projection and Evidence

To quantify the progression of the latent flow, we project the high-dimensional state $\boldsymbol{z}(t)$ into the categorical logit space $\mathbb{R}^C$. We follow the Tuned Lens protocol (Belrose et al., 2023) and learn a collection of layer-specific affine transformations $\Phi_t : \mathbb{R}^d \to \mathbb{R}^C$, where

$$\Phi_t(\boldsymbol{z}) = \mathbf{W}_t \boldsymbol{z} + \mathbf{b}_t. \tag{3}$$

These transformations are trained post hoc on frozen backbones and are used only for analysis; they are not involved

in DTS training. For a sample belonging to class $k$, we define the **Semantic Evidence** $E(z)$ as

$$E(\boldsymbol{z}) = [\Phi_t(\boldsymbol{z})]_k - \frac{1}{C-1} \sum_{j \neq k} [\Phi_t(\boldsymbol{z})]_j. \qquad (4)$$

This scalar function $E : \mathbb{R}^d \to \mathbb{R}$ represents the semantic polarization of a latent state. Since $\Phi_t$ is affine, the semantic gradient $\nabla_{\boldsymbol{z}} E(\boldsymbol{z})$ is a constant vector at each depth, representing a depth-specific direction of categorical refinement in the latent representation space. This observable allows us to reduce the complex high-dimensional trajectory into a scalar-valued progress metric $E(t) = E(\boldsymbol{z}(t))$.

### 3.3. Latent Density Evolution and Semantic Transport

We characterize the collective behavior of these trajectories through the lens of non-autonomous dynamical systems. While each input $\boldsymbol{x}$ follows a deterministic path approximated by Eq. (1), the data distribution $p_{\text{data}}(\boldsymbol{x})$ induces a time-varying marginal density $p_t(\boldsymbol{z})$ in the representation space:

$$p_t(\boldsymbol{z}) = \int_{\mathcal{X}} p_t(\boldsymbol{z} \mid \boldsymbol{x}) p_{\text{data}}(\boldsymbol{x}) d\boldsymbol{x}. \qquad (5)$$

Under the continuous approximation, this density satisfies the continuity equation:

$$\frac{\partial p_t(\boldsymbol{z})}{\partial t} + \nabla \cdot (p_t(\boldsymbol{z}) \boldsymbol{u}(\boldsymbol{z}, t)) = 0, \qquad (6)$$

where $\boldsymbol{u}(\boldsymbol{z}, t)$ is the marginal velocity field. Standard training primarily constrains the boundary condition at the terminal state $t = 1$, leaving the intermediate transient dynamics weakly regulated. To introduce a structured inductive bias, we analyze the evolution of the mean semantic evidence

$$\bar{E}(t) = \int E(\boldsymbol{z}) p_t(\boldsymbol{z}) d\boldsymbol{z}. \qquad (7)$$

By substituting Eq. (6) and applying the transport theorem (Villani, 2008), we obtain the semantic transport relation:

$$\frac{d\bar{E}(t)}{dt} = \mathbb{E}_{p_t} \left[ \boldsymbol{u}(\boldsymbol{z}, t) \cdot \nabla_{\boldsymbol{z}} E(\boldsymbol{z}) \right]. \qquad (8)$$

Equation (8) links the rate of mean evidence growth to the projection of the latent velocity field onto the semantic gradient. It should not be interpreted as proving that a particular evidence schedule globally determines the full high-dimensional vector field. Rather, it shows that controlling the depth-wise progression of semantic evidence provides a tractable way to regularize the task-relevant semantic component of the latent flow. This observation motivates DTS in Sec. 4, where we prescribe a depth-dependent evidence schedule through auxiliary supervision.

### 3.4. Identification of the Exponential Kinematic Prior

To identify the empirical kinematics of the CDF, we probe converged models across diverse scales and tasks. As summarized in Figure 2, our analysis reveals a robust empirical regularity where semantic refinement follows an exponential schedule $E(t) \approx ae^{\beta t} + c$ with $R^2 > 0.97$. This pattern suggests a state-dependent growth law:

$$\frac{dE}{dt} \approx \beta(E - c). \qquad (9)$$

Equation (9) indicates that the velocity of refinement approximately scales with the current evidence level. This exponential signature suggests that residual hierarchies tend to prioritize an initial phase of feature incubation prior to a rapid transition toward categorical certainty. We therefore use this multiplicative progression as an empirically supported kinematic prior for trajectory supervision.

## 4. Deep Trajectory Supervision

### 4.1. Trajectory Alignment Objective

DTS implements trajectory supervision by attaching auxiliary classification heads to intermediate layers. For each layer $l \in \{1, \dots, L\}$ at normalized depth $t = l/L$, we use a classification head $\mathcal{H}_l$ to project the latent state $\boldsymbol{z}_l$ into the logit space:

$$\boldsymbol{s}_l = \mathcal{H}_l(\boldsymbol{z}_l). \qquad (10)$$

The corresponding prediction is

$$\hat{\boldsymbol{p}}_l = \text{Softmax}(\boldsymbol{s}_l). \qquad (11)$$

The goal of DTS is to provide a target confidence that changes with depth. We define a time-dependent target gap

$$G(t) = G_{\max} \alpha(t), \qquad (12)$$

where $G_{\max}$ is the target terminal margin and $\alpha(t) \in [0, 1]$ is a normalized scheduling function satisfying $\alpha(0) = 0$ and $\alpha(1) = 1$. Given the ground-truth class $y$, we construct a target logit vector

$$\boldsymbol{\tau}_t = G(t)\mathbf{e}_y, \qquad (13)$$

where $\mathbf{e}_y$ is the one-hot ground-truth vector. This vector assigns the value $G(t)$ to the correct class and 0 to all other classes. The depth-dependent target distribution is then defined as

$$\boldsymbol{q}_t = \text{Softmax}(\boldsymbol{\tau}_t). \qquad (14)$$

Unlike standard deep supervision, which applies the same hard target at every depth, DTS changes the target confidence as a function of depth.

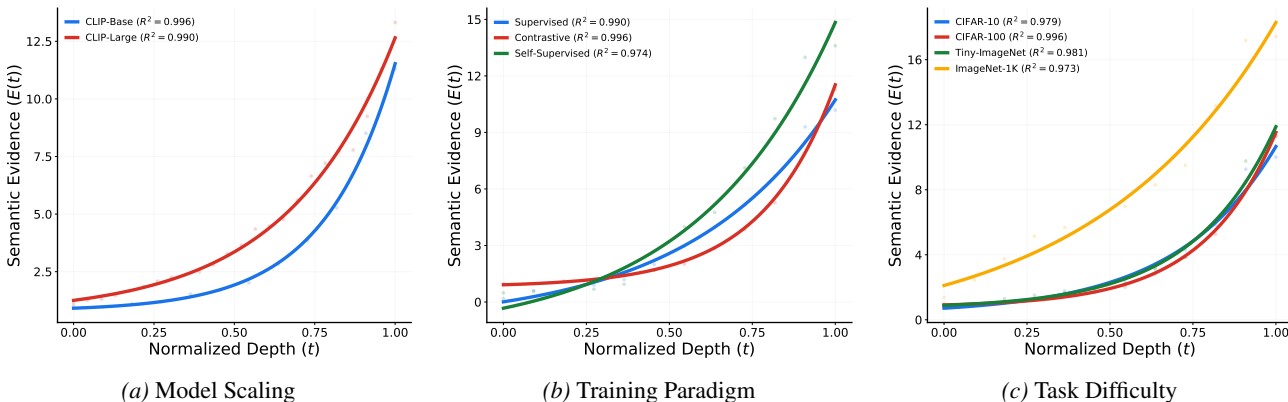

*(a)* Model Scaling          *(b)* Training Paradigm          *(c)* Task Difficulty

*Figure 2.* **Empirical identification of the exponential pattern of semantic evidence accumulation.** We plot the Semantic Evidence $E(t)$ against the normalized depth $t$ to characterize the latent kinematics of inference across various settings. Scatter points represent decoded signals from the Tuned Lens (Belrose et al., 2023); solid lines represent the least-squares exponential fit. **(a) Model Scaling:** Evidence accumulation for CLIP-Base and CLIP-Large models (Radford et al., 2021) on the CIFAR-100 dataset. The exponential curvature remains robust across different parameter scales. **(b) Training Paradigm:** Evaluation of diverse learning objectives on CIFAR-100, including supervised learning (ViT-B), contrastive learning (CLIP-B), and self-supervised learning (DINOv2-B) (Oquab et al., 2023). The kinematic signature remains consistent across different training paradigms. **(c) Task Difficulty:** Progression of semantic refinement using a fixed CLIP-Base backbone across datasets of varying complexity, specifically CIFAR-10, CIFAR-100, Tiny-ImageNet, and ImageNet-1K. The exponential pattern persists across diverse settings, suggesting that it is a stable empirical property of the analyzed ViT representations.

## 4.2. Exponential Supervision Schedule

The schedule $\alpha(t)$ is motivated by the empirical state-dependent growth pattern observed in Sec. 3.4. We use the normalized exponential schedule

$$\alpha(t) = \frac{e^{\beta t} - 1}{e^{\beta} - 1}. \tag{15}$$

The curvature parameter $\beta$ controls how quickly target confidence increases across depth. A larger $\beta$ keeps the early target confidence lower and concentrates the increase of categorical certainty in later layers. This matches the observed pattern in which models first maintain a feature-construction phase and then rapidly increase semantic evidence near the terminal layers. In the limit $\beta \to 0$, the schedule approaches a linear progression. In this work, we use the exponential schedule as an empirically supported prior rather than as a theoretically proven optimum.

## 4.3. Training Objective

The DTS objective combines the layer-wise trajectory alignment loss with the terminal task loss. The local trajectory loss at layer $l$ is

$$\mathcal{L}_l = \mathcal{L}_{\mathrm{CE}}(\hat{\boldsymbol{p}}_l, \boldsymbol{q}_{l/L}). \tag{16}$$

The full training objective is

$$\mathcal{L}_{\mathrm{total}} = \sum_{l=1}^{L} \lambda_l \mathcal{L}_{\mathrm{CE}}(\hat{\boldsymbol{p}}_l, \boldsymbol{q}_{l/L}) + \gamma \mathcal{L}_{\mathrm{task}}(\mathcal{H}_L(\boldsymbol{z}_L), \mathbf{e}_y). \tag{17}$$

Here, $\lambda_l$ is a depth-dependent weighting coefficient, and $\mathcal{L}_{\mathrm{task}}$ denotes the standard hard-label classification loss applied to the final output. The coefficient $\gamma$ controls the contribution of the terminal task loss. In smaller-scale settings, the trajectory term alone can provide sufficient supervision, while in ImageNet-scale training we keep the final task loss active to preserve strong terminal discriminability.

At inference time, all auxiliary heads are removed and predictions are made using only the terminal classifier. Thus, DTS changes the training objective but does not increase inference-time computation.

## 4.4. Geometric Interpretation

We now analyze the local effect of the DTS auxiliary loss. Let the auxiliary head be affine:

$$\mathcal{H}_l(\boldsymbol{z}_l) = \mathbf{W}_l \boldsymbol{z}_l + \boldsymbol{b}_l. \tag{18}$$

The gradient of the local trajectory loss with respect to the latent state is

$$\nabla_{\boldsymbol{z}_l} \mathcal{L}_l = \mathbf{W}_l^{\top}(\hat{\boldsymbol{p}}_l - \boldsymbol{q}_{l/L}). \tag{19}$$

Writing the rows of $\mathbf{W}_l$ as $\boldsymbol{w}_{l,1}, \ldots, \boldsymbol{w}_{l,C}$, the exact form is

$$\nabla_{\boldsymbol{z}_l} \mathcal{L}_l = (\hat{p}_{l,y} - q_{t,y})\boldsymbol{w}_{l,y} + \sum_{j \neq y}(\hat{p}_{l,j} - q_{t,j})\boldsymbol{w}_{l,j}, \quad t = \frac{l}{L}. \tag{20}$$

Because the DTS target assigns the same probability to each non-target class, we have

$$q_{t,j} = \frac{1 - q_{t,y}}{C - 1}, \quad j \neq y. \tag{21}$$

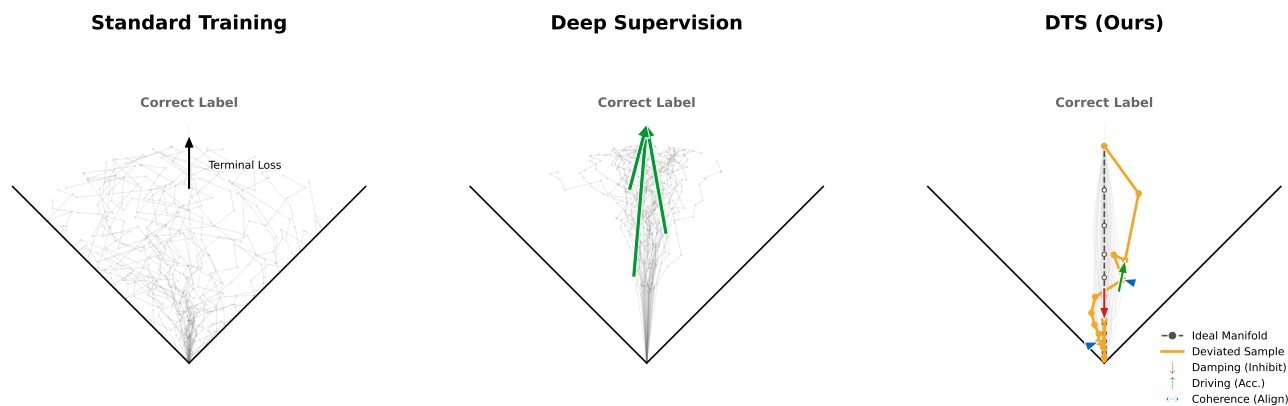

*Figure 3.* **Geometric interpretation of DTS in a polar semantic space.** We visualize the inference process as a discrete semantic trajectory where radius represents Semantic Evidence and angle represents semantic direction. **(Left) Standard Training:** Terminal-only supervision leaves intermediate semantic evolution weakly constrained. **(Middle) Deep Supervision:** Uniform hard-label supervision can impose premature categorical confidence at shallow layers. **(Right) DTS (Ours):** DTS aligns intermediate supervision with the empirically observed exponential evidence schedule. The resulting auxiliary signal encourages a more coherent semantic trajectory by correcting class-specific drift while regulating the depth-wise growth of target confidence.

Define the mean non-target classifier direction as

$$\bar{\boldsymbol{w}}_{l,\neg y} = \frac{1}{C-1} \sum_{j \neq y} \boldsymbol{w}_{l,j}. \qquad (22)$$

Then Eq. (20) can be rewritten in the centered form

$$\nabla_{\boldsymbol{z}_l} \mathcal{L}_l = \left(\hat{p}_{l,y} - q_{t,y}\right) \left(\boldsymbol{w}_{l,y} - \bar{\boldsymbol{w}}_{l,\neg y}\right)$$
$$+ \sum_{j \neq y} \hat{p}_{l,j} \left(\boldsymbol{w}_{l,j} - \bar{\boldsymbol{w}}_{l,\neg y}\right). \qquad (23)$$

The first term controls the semantic margin between the target classifier and the average non-target classifier. When the current target confidence is higher than the scheduled target, this term discourages premature saturation; when the target confidence lags behind the schedule, it encourages further semantic refinement. The second term depends on the model's current distribution over incorrect classes and corrects class-specific semantic drift. This gives a geometric interpretation of DTS as a local semantic regulator. The complete derivation of Eq. (23) is provided in Appendix A.1.

### 4.5. Relation to Standard Deep Supervision

DTS is closely related to deep supervision, but differs in the target assigned to intermediate layers. Standard deep supervision typically applies the same one-hot label at every depth, implicitly asking shallow layers to become as confident as the terminal classifier. DTS instead assigns a depth-dependent soft target. Early layers receive low-confidence targets, while later layers receive increasingly sharper targets according to Eq. (15). Therefore, DTS does not merely add auxiliary losses; it specifies how target confidence should evolve across depth.

This distinction is important because it separates the role of intermediate supervision from the role of trajectory shape. In our experiments, fixed-margin and depth-weighted fixed-margin controls are used to test whether the improvement comes only from auxiliary heads or loss weighting. The results show that the exponential target trajectory itself is an important factor in the observed gains.

## 5. Experiments

We evaluate DTS across datasets with different scales and semantic granularities, including CIFAR-10/100, Tiny-ImageNet, Food-101, iNat19, and ImageNet-1K (Russakovsky et al., 2015). We first report the main training dynamics and ImageNet-scale results, and then analyze whether DTS changes the internal semantic trajectory. We evaluate DTS across datasets with different scales and semantic granularities, including CIFAR-10/100, Tiny-ImageNet, Food-101, iNat19, and ImageNet-1K (Russakovsky et al., 2015). We first report the main training dynamics and ImageNet-scale results, then analyze the internal semantic trajectory, compute-normalized efficiency, and schedule ablations. Additional comparisons across ResNet and ConvNeXt backbones and stronger intermediate-supervision baselines are provided in Appendix B.1.

### 5.1. Experimental Setup

**Datasets.** Our evaluation suite spans coarse classification, medium-scale recognition, and fine-grained recognition. Dataset statistics are summarized in Table 1.

**Architectures.** For small- and mid-scale experiments, we use a standardized ViT-Small backbone unless otherwise

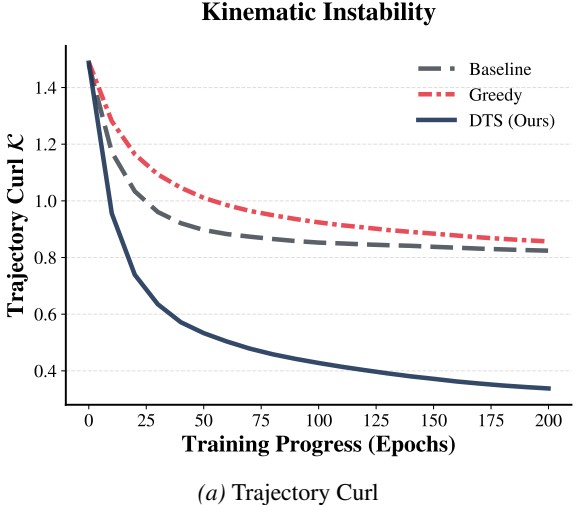

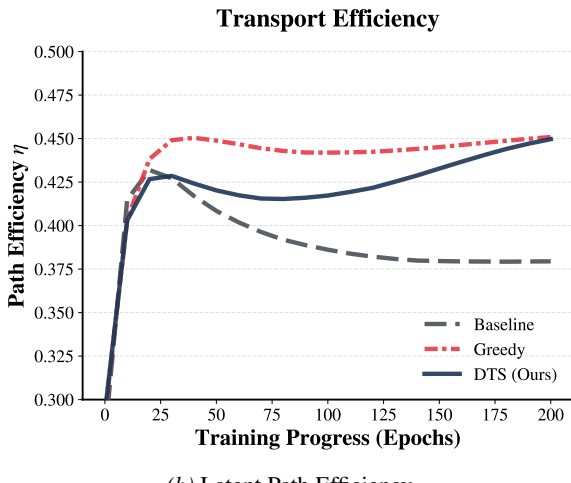

*(a)* Trajectory Curl  ·  *(b)* Latent Path Efficiency

*Figure 4.* **Kinematic evolution of inference flow properties during training.** We monitor the transport characteristics of the conditional discriminative flow on CIFAR-100. **Left:** Trajectory Curl $\mathcal{K}$ quantifies semantic instability. DTS achieves lower curl, indicating a smoother trajectory. **Right:** Latent Path Efficiency $\eta$ measures the ratio of net semantic displacement to total semantic arc length. DTS improves path efficiency while avoiding the premature confidence imposed by hard-label intermediate supervision.

*Table 1.* **Dataset statistics across diverse semantic granularities and scales.** Our evaluation suite spans from low-resolution coarse classification to high-resolution fine-grained recognition tasks.

| Dataset | Classes | Resolution | Train size | Val size |
|---|---|---|---|---|
| CIFAR-10 | 10 | $32^2$ | 50,000 | 10,000 |
| CIFAR-100 | 100 | $32^2$ | 50,000 | 10,000 |
| Tiny-ImageNet | 200 | $64^2$ | 100,000 | 10,000 |
| Food-101 | 101 | $224^2$ | 75,750 | 25,250 |
| iNat19 | 1,010 | $224^2$ | 265,240 | 3,030 |
| ImageNet-1K | 1,000 | $224^2$ | 1,281,167 | 50,000 |

stated. For ImageNet-1K, we incorporate DTS into the DeiT-III training recipe (Touvron et al., 2022). Additional ResNet and ConvNeXt results are reported in Appendix B.1.

**Baselines.** We compare DTS with terminal-only training and standard deep supervision. Additional comparisons with aligned training, fixed-margin supervision, and depth-weighted fixed-margin controls are reported in Appendix B.1. These additional controls isolate the effect of the supervision trajectory from the presence of auxiliary heads and depth-wise loss weighting.

## 5.2. Training Dynamics

Table 2 reports Top-1 accuracy at different stages of training. DTS consistently improves the training trajectory over the terminal-only baseline. The gains are visible not only at the final checkpoint but also during earlier training stages, suggesting that the depth-dependent supervision signal improves optimization dynamics.

Compared with standard deep supervision, DTS avoids imposing the same hard target at all depths. This is particularly important for early layers, where premature categorical confidence may conflict with feature construction. Instead, DTS provides a softer target at shallow depths and gradually increases target confidence toward the terminal layers.

## 5.3. ImageNet-Scale Evaluation

We further evaluate DTS on ImageNet-1K using the competitive DeiT-III recipe. Table 3 shows that DTS provides additive gains on top of this strong training setup. The improvement is modest but consistent, indicating that the proposed trajectory supervision remains useful even when combined with modern regularization and optimization techniques.

## 5.4. Kinematic Analysis

We analyze how DTS changes the internal semantic trajectory. The exponential progression of Semantic Evidence $E(t)$ motivates the target schedule, but the stability of the trajectory can also be characterized by geometric diagnostics.

**Trajectory Curl.** We quantify semantic instability through Trajectory Curl:

$$\mathcal{K} = \int_0^1 \left\| \frac{d}{dt}\left( \frac{\dot{\boldsymbol{y}}(t)}{\|\dot{\boldsymbol{y}}(t)\|_2} \right) \right\|_2 dt, \tag{24}$$

where $\boldsymbol{y}(t)$ denotes the semantic-space trajectory decoded by the Tuned Lens. Lower curl indicates a smoother semantic trajectory.

*Table 2.* **Training dynamics.** All models are trained from scratch unless otherwise specified. We report Top-1 accuracy at intervals of 10 percent total training progress. DTS consistently improves over terminal-only training and standard deep supervision. Bold indicates the best result in each column per group.

| Method | 10% | 20% | 30% | 40% | 50% | 60% | 70% | 80% | 90% | 100% | $\Delta$ |
|---|---|---|---|---|---|---|---|---|---|---|---|
| *CIFAR-10* | | | | | | | | | | | |
| Baseline | 66.6 | 76.3 | 79.5 | 80.3 | 80.5 | 80.5 | 81.4 | 81.5 | 82.1 | 82.2 | – |
| DTS | **72.2** | **79.3** | **80.1** | **80.9** | **82.1** | **81.8** | **82.4** | **83.3** | **83.6** | **83.9** | +1.7 |
| *CIFAR-100* | | | | | | | | | | | |
| Baseline | 47.1 | 50.6 | 51.7 | 51.9 | 52.6 | 53.0 | 53.8 | 54.2 | 55.0 | 54.8 | – |
| Deep Sup. | 46.4 | 50.1 | 52.0 | 52.0 | 53.8 | **54.2** | 54.7 | 55.4 | 55.8 | 55.9 | +1.1 |
| DTS | **47.2** | **52.2** | **53.4** | **53.8** | **53.9** | 53.8 | **55.6** | **56.3** | **56.8** | **57.0** | +2.2 |
| *CIFAR-100 + RandAug* | | | | | | | | | | | |
| Baseline | **48.6** | 57.3 | 59.2 | **60.9** | **61.4** | **61.2** | 62.4 | 63.3 | 63.3 | 63.5 | – |
| Deep Sup. | 42.8 | 54.2 | 57.4 | 59.7 | 60.0 | 60.6 | 61.4 | 62.0 | 62.5 | 62.9 | -0.6 |
| DTS | 47.2 | **58.1** | **59.6** | 60.7 | 61.1 | 61.1 | **62.0** | **63.1** | **63.6** | **64.1** | +0.6 |
| *Tiny-ImageNet* | | | | | | | | | | | |
| Baseline | 36.3 | 35.5 | 35.6 | 36.6 | 37.4 | 38.7 | 38.6 | 39.3 | 39.7 | 39.9 | – |
| DTS | **36.9** | **38.5** | **38.9** | **39.6** | **40.2** | **40.2** | **40.9** | **41.7** | **41.7** | **41.9** | +2.0 |
| *iNat19 + RandAug* | | | | | | | | | | | |
| Baseline | 37.9 | 40.6 | 42.4 | 42.1 | 43.7 | 44.0 | 44.0 | 45.7 | 46.2 | 46.4 | – |
| DTS | **38.8** | **44.4** | **45.1** | **45.5** | **46.0** | **47.2** | **48.8** | **48.6** | **49.0** | **49.1** | +2.7 |

*Table 3.* **ImageNet-1K results with DeiT-III.** DTS improves over the DeiT-III baseline while using the same inference-time architecture.

| Model | Resolution | Baseline | DTS |
|---|---|---|---|
| DeiT-III-T | $224^2$ | 72.7 | **75.2** |
| DeiT-III-S | $224^2$ | 80.4 | **81.2** |
| DeiT-III-B | $192^2$ | 82.3 | **82.6** |

**Latent Path Efficiency.** We measure the directness of the semantic path using Latent Path Efficiency:

$$\eta = \frac{\|\boldsymbol{y}(1) - \boldsymbol{y}(0)\|_2}{\int_0^1 \|\dot{\boldsymbol{y}}(t)\|_2 dt}. \tag{25}$$

A higher value of $\eta$ indicates that the semantic trajectory is more direct in the decoded logit space. The discrete implementation and the definition of $\boldsymbol{y}_l$ are provided in Appendix A.4.

Figure 4 shows that DTS reduces trajectory curl and improves path efficiency compared with terminal-only training. Compared with standard deep supervision, DTS follows a more gradual and stable progression, consistent with its depth-dependent target confidence schedule.

### 5.5. Compute-Normalized Training Efficiency

Since DTS introduces auxiliary heads during training, we further report compute-normalized efficiency in Table 4. The auxiliary heads are lightweight and removed during inference, so DTS does not change the test-time architecture. During training, the added overhead is small: $+3.55\%$ for DeiT-III-S and $+2.14\%$ for DeiT-III-B. Despite this overhead, DTS improves time-to-target accuracy in representative ImageNet-scale settings.

On DeiT-III-S, DTS reaches 60, 70, and 80 Top-1 accuracy faster than the baseline despite the small per-step overhead. On DeiT-III-B, the overhead decreases with model size, and DTS remains comparable or faster in time-to-target accuracy. These results support that the convergence improvement is not merely an epoch-normalized effect.

### 5.6. Ablation Studies

We next ablate the design of the DTS target schedule. First, we compare the proposed exponential trajectory with alternative target confidence schedules on ImageNet-1K using DeiT-III-S. As shown in Table 5, the exponential schedule performs best among the tested variants, suggesting that the target confidence trajectory itself affects optimization and final accuracy.

We also evaluate the sensitivity of DTS to the exponential

*Table 4.* **Compute-normalized training efficiency.** We report per-step wall-clock time, training overhead, FLOPs per sample, peak memory, and time-to-target accuracy. DeiT-III-B results are measured at $192 \times 192$ resolution.

| Metric | DeiT-III-S Baseline | DeiT-III-S DTS | DeiT-III-B Baseline | DeiT-III-B DTS |
|---|---|---|---|---|
| Avg. time / step | 155.72 ms | 161.25 ms | 277.33 ms | 283.27 ms |
| Training overhead | – | +3.55% | – | +2.14% |
| FLOPs / sample | 4.599 G | 4.603 G | 12.789 G | 12.797 G |
| Peak memory | 13.56 GB | 13.65 GB | 18.97 GB | 19.11 GB |
| Time to Acc = 60 | 2.68 h | **1.23 h** | 2.45 h | **2.36 h** |
| Time to Acc = 70 | 5.38 h | **3.42 h** | 4.81 h | **4.48 h** |
| Time to Acc = 80 | 10.00 h | **8.59 h** | 12.22 h | **12.20 h** |

*Table 5.* **Ablation on trajectory shape.** We compare different target confidence schedules on ImageNet-1K using DeiT-III-S.

| Variant | Top-1 Acc. |
|---|---|
| DTS (tanh) | 80.7 |
| DTS (linear) | 80.9 |
| DTS (exponential) | **82.2** |

*Table 6.* **Hyperparameter sensitivity on ImageNet-1K using DeiT-III-S at 50 percent training progress.** Bold indicates the best configuration.

| Configuration | Hyperparameters | Top-1 Acc. |
|---|---|---|
| Linear Trajectory | $G_{\max} = 10$, linear | 73.60 |
| Curvature Default | $G_{\max} = 10, \beta = 0.2$ | 73.48 |
| | $G_{\max} = 10, \beta = 0.5$ | **73.83** |
| | $G_{\max} = 10, \beta = 2.0$ | 73.78 |
| | $G_{\max} = 10, \beta = 4.0$ | 73.57 |
| Target Margin | $G_{\max} = 8, \beta = 0.5$ | 73.49 |
| | $G_{\max} = 12, \beta = 0.5$ | 73.82 |

curvature $\beta$ and the target terminal margin $G_{\max}$. Table 6 reports the ablation results on ImageNet-1K using DeiT-III-S at 50 percent training progress. DTS is robust across a moderate range of $\beta$, while increasing $G_{\max}$ beyond 10 yields diminishing returns. This suggests that the main effect comes from the depth-wise shape of the schedule rather than from simply increasing the target logit magnitude.

## 6. Conclusion

In this work, we formalized the forward pass of deep classifiers as a **Conditional Discriminative Flow** and studied the depth-wise progression of semantic evidence. Using Tuned Lens analysis, we observed a robust approximately exponential pattern of evidence accumulation across model scales, training paradigms, and datasets. Motivated by this empirical regularity, we proposed **Deep Trajectory Supervision**, a simple extension of deep supervision that assigns depth-dependent soft targets to intermediate layers. Across ViTs, ResNets, and ConvNeXts, DTS improves terminal performance and compares favorably with standard deep supervision, aligned training, and fixed-margin controls. These results suggest that how target confidence evolves with depth is an important design choice for intermediate supervision.

## 7. Limitations

DTS is motivated by a continuous-depth interpretation of residual architectures, but practical networks are finite-depth discrete systems. We therefore use the CDF formulation as an interpretive approximation rather than as a claim of exact continuum-limit dynamics. Appendix A.2 provides the corresponding discrete residual approximation. In addition, the exponential schedule should be interpreted as an empirically supported trajectory prior, not as a proof of global optimality. Finally, DTS introduces auxiliary heads during training. These heads are removed at inference time, and the compute-normalized measurements in Sec. 5.5 show that the additional training overhead is small.

## Impact Statement

The societal impact of this work is indirect and depends on the downstream use of image classification systems. Improved visual recognition models can support beneficial applications such as scientific image analysis, environmental monitoring, and assistive technologies. Our method does not introduce new data collection, personal information processing, or deployment-specific mechanisms, but it could be applied to models used in such settings.

## Acknowledgements

This work was supported by the FIS project GUIDANCE (No. FIS2023-03251) and the EU Horizon project EL-LIOT (No. 101214398). The authors gratefully acknowledge the Gauss Centre for Supercomputing e.V. (www.gauss-centre.eu) for supporting this project by providing computing time on the GCS Supercomputer JUWELS at Jülich Supercomputing Centre (JSC).

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

# A. Additional Theoretical Details

This section provides additional details for the theoretical formulation used in the main paper. We first derive the exact local gradient induced by the DTS auxiliary loss. We then clarify how the continuous Conditional Discriminative Flow (CDF) view relates to finite-depth residual networks. Finally, we provide details on the Tuned Lens protocol, the semantic-space diagnostics, and the intended scope of the exponential supervision schedule.

## A.1. Full Derivation of the DTS Gradient

We derive the local gradient induced by the layer-wise trajectory alignment loss. For a layer $l$ at normalized depth $t = l/L$, let the auxiliary classifier be an affine head

$$\mathcal{H}_l(\boldsymbol{z}_l) = \boldsymbol{s}_l = \mathbf{W}_l \boldsymbol{z}_l + \boldsymbol{b}_l, \tag{26}$$

where $\boldsymbol{s}_l \in \mathbb{R}^C$ denotes the auxiliary logits. The predicted distribution is

$$\hat{\boldsymbol{p}}_l = \mathrm{Softmax}(\boldsymbol{s}_l). \tag{27}$$

DTS constructs a depth-dependent target distribution $\boldsymbol{q}_t$ from the target logit vector

$$\boldsymbol{\tau}_t = G(t)\mathbf{e}_y, \tag{28}$$

where $y$ is the ground-truth class and $G(t) = G_{\max}\alpha(t)$ is the target evidence gap. The target distribution is therefore

$$\boldsymbol{q}_t = \mathrm{Softmax}(\boldsymbol{\tau}_t). \tag{29}$$

Since only the ground-truth logit is assigned a nonzero value in $\boldsymbol{\tau}_t$, the target probabilities are

$$q_{t,y} = \frac{e^{G(t)}}{e^{G(t)} + C - 1}, \qquad q_{t,j} = \frac{1}{e^{G(t)} + C - 1}, \quad j \neq y. \tag{30}$$

The local trajectory alignment loss at layer $l$ is the cross-entropy between the target distribution and the auxiliary prediction:

$$\mathcal{L}_l = \mathcal{L}_{\mathrm{CE}}(\hat{\boldsymbol{p}}_l, \boldsymbol{q}_t) = -\sum_{j=1}^{C} q_{t,j} \log \hat{p}_{l,j}. \tag{31}$$

For the softmax cross-entropy loss, the derivative with respect to the auxiliary logits is

$$\frac{\partial \mathcal{L}_l}{\partial \boldsymbol{s}_l} = \hat{\boldsymbol{p}}_l - \boldsymbol{q}_t. \tag{32}$$

Applying the chain rule through the affine head gives

$$\nabla_{\boldsymbol{z}_l} \mathcal{L}_l = \mathbf{W}_l^\top (\hat{\boldsymbol{p}}_l - \boldsymbol{q}_t). \tag{33}$$

Writing $\mathbf{W}_l$ row-wise as

$$\mathbf{W}_l = \begin{bmatrix} \boldsymbol{w}_{l,1}^\top \\ \cdots \\ \boldsymbol{w}_{l,C}^\top \end{bmatrix}, \tag{34}$$

Eq. (33) becomes

$$\nabla_{\boldsymbol{z}_l} \mathcal{L}_l = (\hat{p}_{l,y} - q_{t,y})\boldsymbol{w}_{l,y} + \sum_{j \neq y} (\hat{p}_{l,j} - q_{t,j})\boldsymbol{w}_{l,j}. \tag{35}$$

This is the exact local gradient form. Since the DTS target assigns the same probability to every non-target class, all non-target classes share

$$q_{t,j} = \frac{1 - q_{t,y}}{C - 1}, \quad j \neq y. \tag{36}$$

Define the mean non-target classifier direction as

$$\bar{\boldsymbol{w}}_{l,\neg y} = \frac{1}{C-1} \sum_{j\neq y} \boldsymbol{w}_{l,j}. \tag{37}$$

Then Eq. (35) can be rewritten as

$$\nabla_{\boldsymbol{z}_l}\mathcal{L}_l = (\hat{p}_{l,y} - q_{t,y})\boldsymbol{w}_{l,y} + \sum_{j\neq y}\hat{p}_{l,j}\boldsymbol{w}_{l,j} - (1 - q_{t,y})\bar{\boldsymbol{w}}_{l,\neg y}$$

$$= (\hat{p}_{l,y} - q_{t,y})\left(\boldsymbol{w}_{l,y} - \bar{\boldsymbol{w}}_{l,\neg y}\right) + \sum_{j\neq y}\hat{p}_{l,j}\left(\boldsymbol{w}_{l,j} - \bar{\boldsymbol{w}}_{l,\neg y}\right). \tag{38}$$

This centered form separates the target-class correction from the prediction-dependent non-target correction. The first term adjusts the representation along the semantic margin direction between the target classifier and the average non-target classifier. The second term depends on the model's current distribution over incorrect classes and therefore corrects class-specific semantic drift.

The derivation above analyzes the direct local gradient produced by the auxiliary DTS loss. The gradient of the full training objective additionally includes the weighting coefficient $\lambda_l$ and, for shared parameters, the usual backpropagated contributions from deeper auxiliary losses and the terminal task loss:

$$\mathcal{L}_{\text{total}} = \sum_{l=1}^{L}\lambda_l\mathcal{L}_{\text{CE}}(\hat{\boldsymbol{p}}_l, \boldsymbol{q}_{l/L}) + \gamma\mathcal{L}_{\text{task}}(\mathcal{H}_L(\boldsymbol{z}_L), \mathbf{e}_y). \tag{39}$$

Therefore, Eq. (38) should be understood as the local mechanism of the DTS auxiliary signal, rather than as the complete gradient of all model parameters.

## A.2. Discrete Residual Approximation and Semantic Transport

The CDF formulation in the main text uses a continuous-time view to describe the layer-wise evolution of residual architectures. Practical networks, however, are finite-depth discrete systems. This section clarifies the approximation.

Consider a residual network with normalized step size $\Delta t = 1/L$. Its layer-wise update can be written as

$$\boldsymbol{z}_{l+1} = \boldsymbol{z}_l + \mathcal{F}_l(\boldsymbol{z}_l, \boldsymbol{x}), \qquad t_l = \frac{l}{L}. \tag{40}$$

The continuous CDF view corresponds to the first-order approximation

$$\mathcal{F}_l(\boldsymbol{z}_l, \boldsymbol{x}) = \Delta t\,\boldsymbol{v}(\boldsymbol{z}_l, t_l \mid \boldsymbol{x}) + \mathcal{O}(\Delta t^2), \tag{41}$$

where $\boldsymbol{v}$ is the corresponding input-conditioned velocity field.

Let $E$ be a smooth semantic observable. A first-order Taylor expansion around $\boldsymbol{z}_l$ gives

$$E(\boldsymbol{z}_{l+1}) - E(\boldsymbol{z}_l) = \nabla_{\boldsymbol{z}}E(\boldsymbol{z}_l)^\top (\boldsymbol{z}_{l+1} - \boldsymbol{z}_l) + \mathcal{O}\big(\|\boldsymbol{z}_{l+1} - \boldsymbol{z}_l\|_2^2\big)$$

$$= \nabla_{\boldsymbol{z}}E(\boldsymbol{z}_l)^\top \mathcal{F}_l(\boldsymbol{z}_l, \boldsymbol{x}) + \mathcal{O}\big(\|\mathcal{F}_l(\boldsymbol{z}_l, \boldsymbol{x})\|_2^2\big)$$

$$= \Delta t\,\nabla_{\boldsymbol{z}}E(\boldsymbol{z}_l)^\top \boldsymbol{v}(\boldsymbol{z}_l, t_l \mid \boldsymbol{x}) + \mathcal{O}(\Delta t^2). \tag{42}$$

Dividing both sides by $\Delta t$ yields the discrete analogue of the semantic transport relation:

$$\frac{E(\boldsymbol{z}_{l+1}) - E(\boldsymbol{z}_l)}{\Delta t} = \nabla_{\boldsymbol{z}}E(\boldsymbol{z}_l)^\top \boldsymbol{v}(\boldsymbol{z}_l, t_l \mid \boldsymbol{x}) + \mathcal{O}(\Delta t). \tag{43}$$

Taking expectation over the data-induced representation distribution gives

$$\frac{\bar{E}_{l+1} - \bar{E}_l}{\Delta t} = \mathbb{E}_{p_{t_l}}\left[\boldsymbol{u}(\boldsymbol{z}, t_l) \cdot \nabla_{\boldsymbol{z}}E(\boldsymbol{z})\right] + \mathcal{O}(\Delta t), \tag{44}$$

where $\bar{E}_l = \mathbb{E}_{p_{t_l}}[E(\boldsymbol{z})]$ and $\boldsymbol{u}$ denotes the marginal velocity field. In the limit $\Delta t \to 0$, this recovers the continuous relation used in Eq. (8).

This derivation clarifies the intended scope of the CDF formulation. The continuous equation is not used as a claim that finite-depth neural networks exactly satisfy a continuum-limit ODE. Rather, it provides a first-order lens for interpreting how residual updates change semantic evidence. The resulting transport relation shows that evidence growth is related to the projection of the latent velocity onto the semantic gradient. It does not prove that any particular evidence schedule is globally optimal, nor that matching a scalar evidence target fully determines the high-dimensional vector field.

### A.3. Tuned Lens Protocol

We use Tuned Lens probes only for analysis. For each layer $l$, we freeze the backbone and train a layer-specific affine map

$$\Phi_l(\boldsymbol{z}_l) = \mathbf{W}_l^{\Phi} \boldsymbol{z}_l + \boldsymbol{b}_l^{\Phi} \tag{45}$$

to decode the ground-truth label from the intermediate representation $\boldsymbol{z}_l$. The backbone parameters are not updated when training these probes. The probes are trained post hoc on the training split and are not used in the DTS training objective.

After training the probes, we compute Semantic Evidence by applying $\Phi_l$ to the intermediate representations:

$$E_l(\boldsymbol{z}_l) = [\Phi_l(\boldsymbol{z}_l)]_y - \frac{1}{C-1} \sum_{j \neq y} [\Phi_l(\boldsymbol{z}_l)]_j. \tag{46}$$

This provides a depth-wise scalar estimate of how much class-discriminative information is linearly decodable from each layer. The evidence curves reported in the main text are obtained by averaging this quantity over samples at each depth.

For ViT-style architectures, $\boldsymbol{z}_l$ denotes the hidden state of the [CLS] token. For CNN-style architectures such as ResNets and ConvNeXts, $\boldsymbol{z}_l$ denotes the block-wise or stage-wise representation after global pooling, matching the representation used by the corresponding auxiliary classifier. This makes the analysis applicable beyond the [CLS] token setting.

### A.4. Semantic-Space Diagnostics

The diagnostic metrics in Sec. 5.4 are computed in the decoded semantic space rather than in the raw hidden representation space. Specifically, we define

$$\boldsymbol{y}_l = \Phi_l(\boldsymbol{z}_l) \in \mathbb{R}^C, \tag{47}$$

where $\Phi_l$ is the layer-specific Tuned Lens projection. Thus, $\boldsymbol{y}_l$ represents the semantic-space state of the model at layer $l$.

The discrete version of Latent Path Efficiency is computed as

$$\eta = \frac{\|\boldsymbol{y}_L - \boldsymbol{y}_0\|_2}{\sum_{l=0}^{L-1} \|\boldsymbol{y}_{l+1} - \boldsymbol{y}_l\|_2}. \tag{48}$$

This metric compares the net semantic displacement with the total semantic arc length. A higher value indicates that the semantic trajectory is more direct in the decoded logit space. Since this diagnostic is computed after projecting representations into the same semantic coordinate system, it should be interpreted as a measure of semantic-space path efficiency, not as a claim about the full geometry of the raw hidden states.

### A.5. Scope of the Exponential Schedule

The exponential schedule used by DTS is motivated by the empirical evidence progression observed in converged models. It should be interpreted as an empirically supported trajectory prior rather than as a theoretically proven optimum. The role of DTS is to impose a depth-dependent target confidence that matches this observed progression:

$$\alpha(t) = \frac{e^{\beta t} - 1}{e^{\beta} - 1}. \tag{49}$$

This schedule regulates the target probability distribution $\boldsymbol{q}_t$ used by each auxiliary loss. Although the target evidence value is scalar, it induces a full distribution over classes through $\boldsymbol{q}_t = \mathrm{Softmax}(G(t)\boldsymbol{e}_y)$. The corresponding gradient acts in the hidden representation space through the auxiliary classifier matrix, as shown in Appendix A.1. Therefore, DTS regularizes the task-relevant semantic component of the latent flow, but it does not fully specify the entire high-dimensional trajectory.

# B. Additional Experimental Results

This section reports additional experiments and ablations that complement the main results. This section reports additional experiments that complement the main results. These results clarify the architectural scope of DTS and compare against stronger intermediate-supervision baselines.

## B.1. Expanded Comparison Across Architectures and Baselines

Table 7 provides the expanded comparison across ViT, ResNet, and ConvNeXt backbones. We compare terminal-only training with standard deep supervision (DS), aligned training, fixed-margin supervision (FixMgn), fixed-margin supervision with the same depth-dependent loss weighting as DTS (FixMgn+DW), and DTS. The FixMgn+DW baseline controls for the effect of layer-wise loss coefficients, while keeping the target confidence fixed across depth. The gap between FixMgn+DW and DTS therefore isolates the effect of the supervision trajectory shape.

*Table 7.* **Expanded comparison across architectures and intermediate-supervision baselines.** We report Top-1 accuracy. DTS improves over the baseline across all reported settings and is competitive with or better than standard intermediate-supervision variants.

| Dataset | Backbone | Base | DS | Aligned | FixMgn | FixMgn+DW | DTS |
|---|---|---|---|---|---|---|---|
| CIFAR-10 | ViT-S | 82.2 | 82.2 | 82.9 | 81.9 | 83.0 | **83.9** |
| CIFAR-10 | RN18 | 91.9 | 92.5 | 93.0 | 92.2 | 93.0 | **93.2** |
| CIFAR-10 | RN50 | 91.1 | 92.8 | 93.2 | 93.1 | **94.6** | **94.6** |
| CIFAR-100 + RA | ViT-S | 63.5 | 62.9 | 63.2 | 62.3 | 63.8 | **64.2** |
| CIFAR-100 + RA | RN18 | 71.6 | 73.5 | 73.5 | 72.8 | 74.0 | **74.3** |
| CIFAR-100 + RA | RN50 | 70.2 | 75.4 | **75.6** | 75.2 | 75.3 | **75.6** |
| Tiny-ImageNet | ViT-S | 39.9 | 41.0 | 41.6 | 41.1 | 41.5 | **41.9** |
| Tiny-ImageNet | RN18 | 57.7 | 57.5 | 57.9 | 56.9 | 57.9 | **60.8** |
| Tiny-ImageNet | RN50 | 58.4 | 61.4 | 62.5 | 61.4 | 62.6 | **64.7** |
| Food-101 | ViT-S | 51.1 | 53.3 | 51.4 | 51.2 | 52.9 | **54.8** |
| Food-101 | RN18 | 77.7 | 76.4 | 77.9 | 75.8 | 77.3 | **79.0** |
| Food-101 | RN50 | 78.3 | 80.8 | 81.6 | 80.3 | 81.2 | **82.3** |
| iNat19 | ViT-S | 46.4 | 46.5 | 47.2 | 47.2 | 48.0 | **49.1** |
| iNat19 | RN18 | 57.2 | 60.1 | **61.8** | 59.2 | 59.8 | **61.8** |
| iNat19 | RN50 | 60.1 | 67.1 | 67.4 | 65.1 | 67.9 | **68.7** |
| ImageNet-1K | DeiT-III-S | 80.4 | 80.4 | 77.2 | 80.4 | 80.4 | **81.2** |
| ImageNet-1K | ConvNeXt-T | 82.1 | 82.0 | 79.7 | 81.9 | 82.0 | **82.3** |

The expanded comparison supports two observations. First, DTS is not tied to the `[CLS]` token abstraction of ViTs, since it also improves ResNet and ConvNeXt models. Second, the improvement over FixMgn+DW suggests that the depth-wise target trajectory, rather than only the presence of auxiliary heads or depth-dependent loss weights, is an important contributor to the final performance.

# C. Experimental Configurations

This section provides implementation details required for reproducibility. Our training and evaluation process is executed on NVIDIA A100 GPUs.

## C.1. DTS-Specific Hyperparameters

The core of DTS lies in the exponential evidence schedule and the auxiliary projection heads. We use a standardized configuration across benchmarks unless stated otherwise. The trajectory targets are computed using the normalized exponential function defined in Eq. (15).

The task loss scale $\gamma$ follows the setup used in the main experiments. For smaller and more data-limited benchmarks,

*Table 8.* **Default hyperparameters for Deep Trajectory Supervision.** These values are kept fixed across datasets unless stated otherwise. For BCE-based DeiT-III training, we reduce the trajectory loss scale to match the smaller numerical scale of the BCE objective.

| Category | Hyperparameter | DeiT-III + CE | DeiT-III + BCE | Others |
|---|---|---|---|---|
| Schedule | Target Terminal Margin $G_{\max}$ | 10.0 | 10.0 | 10.0 |
| | Exponential Curvature $\beta$ | 0.5 | 0.5 | 0.5 |
| Auxiliary Heads | Depth Interval | Every Layer | Every Layer | Every Layer |
| Optimization | Initial Trajectory Loss Scale $\lambda_l$ | 0.5/L | 0.001/L | 1.0/L |
| | Linear Decay of Trajectory Loss Scale | True | True | False |
| | Task Loss Scale $\gamma$ | 1 | 1 | 0 |

the trajectory loss alone provides sufficient supervision for stable convergence. For ImageNet-1K, we keep the final task loss active because it preserves strong terminal discriminability while allowing the DTS term to regularize intermediate representations.

### C.2. Training Recipes for Scratch Training

For CIFAR-10/100, Tiny-ImageNet, Food-101, and iNat19, we use a standardized ViT-Small backbone unless otherwise stated. The model consists of 12 Transformer blocks with a hidden dimension of 384 and 6 attention heads.

*Table 9.* **Detailed training recipe for small- to mid-scale benchmarks.** We use a conservative augmentation strategy to isolate the effect of trajectory supervision.

| Hyperparameter | Setting |
|---|---|
| Optimizer | AdamW (Kingma & Ba, 2015) |
| Base Learning Rate | $5 \times 10^{-4}$ |
| Weight Decay | 0.05 |
| Batch Size | 1024 |
| Learning Rate Schedule | Cosine Decay |
| Total Epochs | 200 |
| Default Augmentation | RandomFlip, RandomResizedCrop |
| Additional Augmentation | RandAugment where specified (Cubuk et al., 2020) |

For ResNet and ConvNeXt experiments, we attach auxiliary classification heads to the corresponding block-wise or stage-wise representations. The auxiliary heads are used only during training and are removed at inference time.

### C.3. DeiT-III Protocol for ImageNet-1K

For our large-scale evaluation on ImageNet-1K, we follow the DeiT-III training protocol (Touvron et al., 2022). Following the DeiT-III setting, we train at the prescribed resolution and epoch budget, and evaluate whether DTS provides additive improvements to a strong training recipe. We incorporate the trajectory loss as an auxiliary term without modifying the original regularization stack, which includes stochastic depth, label smoothing, and LayerScale. All DTS-specific parameters, namely $G_{\max}$ and $\beta$, are kept identical to those used in smaller benchmarks.

For DeiT-III-B, we additionally report the $192 \times 192$ setting in the compute-normalized analysis. The auxiliary DTS heads are active only during training and are discarded during inference, so DTS does not increase inference-time FLOPs or memory.

### C.4. Auxiliary Head Implementation

Each auxiliary head consists of a lightweight normalization layer followed by a linear classifier. For a hidden representation $z_l$, the auxiliary head computes

$$s_l = \mathcal{H}_l(z_l). \tag{50}$$

For ViT-style models, $z_l$ is the [CLS] token at layer $l$. For CNN-style models, $z_l$ is obtained by global pooling the corresponding intermediate feature map. The auxiliary heads are optimized jointly with the backbone during training. At test time, all auxiliary heads are removed and predictions are made using only the terminal classifier.

