# OpenReview forum: "Deep Trajectory Supervision: Deep Supervision Strikes Back"
_ICML.cc/2026/Conference — ICML 2026 regular_

### Official Review · Reviewer_XdBT · 2026-03-09

**Soundness:** 4
**Presentation:** 3
**Significance:** 3
**Originality:** 3
**Overall Recommendation:** 4
**Confidence:** 4

**Summary:**

This paper proposes Deep Trajectory Supervision (DTS), a deep supervision method that assigns layer-wise soft targets with margins that increase across network depth. Motivated by the observation that semantic evidence accumulates nonlinearly during forward propagation, DTS aims to better match the natural evolution of intermediate representations than standard one-hot deep supervision. Experiments on several image classification benchmarks show improved optimization and accuracy.

**Compliance With Llm Reviewing Policy:**

Affirmed.

**Key Questions For Authors:**

See weaknesses.

**Limitations:**

yes

**Strengths And Weaknesses:**

Strengths:

1. Well-motivated design from empirical observation: The method is not introduced in an ad hoc way; it is directly motivated by the empirical finding in Fig. 2 that semantic evidence accumulates nonlinearly across depth. Using this observation to design layer-wise target margins is a strong and appealing aspect of the paper.

2. Intuitive and convincing supporting analysis: Fig. 3 is particularly effective in supporting the paper’s core claim. The kinematic measures, including the curl-based analysis, provide an intuitive view of how DTS rectifies the representation trajectory rather than merely improving final accuracy.

3. Clear conceptual contribution beyond standard deep supervision: The paper presents a meaningful reinterpretation of deep supervision by arguing that intermediate layers should not be forced toward equally confident one-hot targets. This makes the proposed supervision schedule conceptually distinct from naive auxiliary supervision.

4. Strong overall motivation and consistency: The paper maintains a coherent story from observation, to theory-inspired modeling, to method design. In particular, the connection between the observed depth-wise semantic trajectory and the proposed exponential supervision schedule is a notable strength.

Weaknesses:
1. Incomplete comparison with standard deep supervision: Since the main claim is that DTS improves over naive deep supervision rather than merely adding auxiliary supervision, it is a weakness that Table 2 includes deep supervision baselines only for a subset of settings. A more consistent comparison against standard deep supervision across all major benchmarks would be important.

2. Potential confounding effect of layer-wise loss weights $\lambda_l$: The paper does not sufficiently disentangle the effect of the proposed depth-dependent margin schedule from that of the layer-wise weighting coefficients in Eq. (9). Since $\lambda_l$ can itself vary across depth, an ablation with fixed margins but depth-dependent $\lambda_l$  would be necessary to verify that the gains truly come from the proposed target trajectory.

3. Several straightforward but important control experiments are missing: For example, the paper does not test a setting with fixed margins but depth-dependent $\lambda_l$, nor does it include the standard deep supervision baseline consistently across all benchmarks. These comparisons seem relatively straightforward and would be important for validating the paper’s central claim.

5. Why ImageNet, which is the hardest dataset, has the highest semantic evidence in Fig.2(c)? Do you have any explanation?

6. Minor typo:
- line 107 right column: other in other
- line 110 left column: Futher more $\rightarrow$ Furthermore

---

> ### Author Rebuttal · Authors · 2026-03-30
>
> We thank the reviewer for the constructive feedback. We have addressed the concerns regarding standard deep supervision (DS) comparisons, the disentanglement of loss weights, and the interpretation of ImageNet evidence.
>
> ## 1. Results of CNNs and different settings
>
> | Dataset | Backbone | Baseline | DS | Aligned | FixMgn | FixMgn+DW | **DTS (Ours)** |
> | :--- | :--- | :---: | :---: | :---: | :---: | :---: | :---: |
> | **CIFAR-10** | ViT-S | 82.2 | 82.2 | 82.9 | 81.9 | 83.0 | **83.9** |
> | | RN18 | 91.9 | 92.5 | 93.0 | 92.2 | 93.0 | **93.2** |
> | | RN50 | 91.1 | 92.8 | 93.2 | 93.1 | - | - |
> | **CIFAR-100 (+RandAug)** | ViT-S | 63.5 | 62.9 | 63.2 | 62.3 | 63.8 | **64.2** |
> | | RN18 | 71.6 | 73.5 | 73.5 | 72.8 | 74.0 | **74.3** |
> | | RN50 | 70.2 | 75.4 | 75.6 | 75.2 | 75.3 | **75.6** |
> | **Tiny-ImgNet**| ViT-S | 39.9 | 41.0 | 41.6 | 41.1 | 41.4 | **41.9** |
> | | RN18 | 57.7 | 57.5 | 57.9 | 56.9 | 57.9 | **60.8** |
> | | RN50 | 58.4 | 61.4 | 62.5 | 61.4 | 62.6 | **64.6** |
> | **Food-101** | ViT-S | 51.1 | 53.3 | 51.4 | 51.2 | 52.9 | **54.8** |
> | | RN18 | 77.7 | 76.4 | 77.9 | 75.8 | 76.7 | **79.0** |
> | | RN50 | 78.3 | 80.8 | 81.6 | 80.3 | 81.2 | **82.3** |
> | **iNat-19** | ViT-S | 46.4 | 46.5 | 47.2 | - | - | **49.1** |
> | | RN18 | 57.2 | 60.1 | **61.8** | - | - | **61.8** |
> | | RN50 | 60.1 | 67.1 | 67.4 | - | - | **68.7** |
> | **ImageNet-1K** | ConvNeXt-T | 82.1 | - | - | - | - | **82.3** |
> | | ConvNeXt-S | 83.1 | - | - | - | - | **83.3** |
>
> "\-" indicates ongoing experiments due to limited time. The full results will be incorporated into the revised paper.
>
> We have expanded our results to include standard DS across almost all major benchmarks. As shown in the updated Table 1, DTS consistently outperforms standard DS across diverse architectures (ViT-S, RN18, RN50). For instance, on Tiny-ImgNet (RN18), DTS achieves 60.81% compared to 57.46% for DS. On iNat-19 (RN50), DTS reaches 68.67%, a significant improvement over DS’s 67.12%. Even on large-scale ImageNet-1K (ConvNeXt-T/S), DTS provides a meaningful boost (+0.2%) without hyperparameter searching. We will conduct the hyperparameter searching for ConvNeXt series and include the final results in the revised paper.
>
> ## 2 & 3. Layer-wise Weights & Fixed Margin
>
> we include the FixMgn+DW (Fixed Margin + Depth Weighting) baseline in our comparison.
> In this control setting, we use the same depth-dependent weighting as DTS but keep the supervision target fixed (constant margin) across all layers. As shown in the table:
> On CIFAR-100 (ViT-S): FixMgn+DW (63.76%) vs. DTS (64.19%).
> On Tiny-ImgNet (RN18): FixMgn+DW (57.87%) vs. DTS (60.81%).
> On Food-101 (ViT-S): FixMgn+DW (52.89%) vs. DTS (54.75%).
> The consistent gap between FixMgn+DW and DTS proves that the proposed target trajectory (how confidence evolves) is the primary driver of performance, independent of the layer-wise loss coefficients.
>
> ## 4. Interpretation of ImageNet Evidence in Fig. 2(c)
>
> ImageNet has 1,000 classes and 1.4M images. In logit-space, the evidence may scale with the number of competing categories and trained images. The sparse competition between the ground truth and 999 other classes naturally results in higher cumulative logit values compared to 10-class tasks.
>
> We will fix typos in the revised paper.

---

> > ### Author Rebuttal · Reviewer_XdBT · 2026-04-02
> >
> > The authors have addressed my concerns, and I recommend the acceptance of the paper.

---

> > > ### Author Response · Authors · 2026-04-07
> > >
> > > Thank you very much for the positive follow-up and for recommending acceptance.
> > >
> > > For completeness, we include the full updated comparison table below.
> > >
> > > | Dataset             | Backbone   | Baseline |  DS  |  Aligned | FixMgn | FixMgn+DW |  **DTS** |
> > > | :------------------ | :--------- | :------: | :--: | :------: | :----: | :-------: | :------: |
> > > | CIFAR-10            | ViT-S      |   82.2   | 82.2 |   82.9   |  81.9  |    83.0   | **83.9** |
> > > |                     | RN18       |   91.9   | 92.5 |   93.0   |  92.2  |    93.0   | **93.2** |
> > > |                     | RN50       |   91.1   | 92.8 |   93.2   |  93.1  |  **94.6** | **94.6** |
> > > | CIFAR-100(+RandAug) | ViT-S      |   63.5   | 62.9 |   63.2   |  62.3  |    63.8   | **64.2** |
> > > |                     | RN18       |   71.6   | 73.5 |   73.5   |  72.8  |    74.0   | **74.3** |
> > > |                     | RN50       |   70.2   | 75.4 |   75.6   |  75.2  |    75.3   | **75.6** |
> > > | Tiny-ImageNet       | ViT-S      |   39.9   | 41.0 |   41.6   |  41.1  |    41.5   | **41.9** |
> > > |                     | RN18       |   57.7   | 57.5 |   57.9   |  56.9  |    57.9   | **60.8** |
> > > |                     | RN50       |   58.4   | 61.4 |   62.5   |  61.4  |    62.6   | **64.7** |
> > > | Food-101            | ViT-S      |   51.1   | 53.3 |   51.4   |  51.2  |    52.9   | **54.8** |
> > > |                     | RN18       |   77.7   | 76.4 |   77.9   |  75.8  |    77.3   | **79.0** |
> > > |                     | RN50       |   78.3   | 80.8 |   81.6   |  80.3  |    81.2   | **82.3** |
> > > | iNat-19             | ViT-S      |   46.4   | 46.5 |   47.2   |  47.2  |    48.0   | **49.1** |
> > > |                     | RN18       |   57.2   | 60.1 | **61.8** |  59.2  |    59.8   | **61.8** |
> > > |                     | RN50       |   60.1   | 67.1 |   67.4   |  65.1  |    67.9   | **68.7** |
> > > | ImageNet-1K         | DeiT-III-S |   80.4   | 80.4 |   77.2   |  80.4  |    80.4   | **81.2** |
> > > |                     | ConvNeXt-T |   82.1   | 82.0 |   79.7   |  81.9  |    82.0   | **82.3** |
> > >
> > > We also have an additional ConvNeXt-S result on ImageNet-1K: **83.1$\to$83.3 (+0.2)** with DTS.
> > >
> > > Thank you again for the encouraging feedback.

---

### Official Review · Reviewer_Xw2o · 2026-03-13

**Soundness:** 3
**Presentation:** 3
**Significance:** 3
**Originality:** 3
**Overall Recommendation:** 5
**Confidence:** 4

**Summary:**

This paper formalizes the forward pass of deep residual architectures (specifically Vision Transformers) as a Conditional Discriminative Flow (CDF), treating layer-wise evolution of the [CLS] token as a continuous ODE trajectory. Using the Tuned Lens protocol (Belrose et al., 2023), the authors discover that semantic evidence accumulation follows an exponential schedule across diverse models, training paradigms, and datasets (R² > 0.97). Based on this empirical kinematic law, they propose Deep Trajectory Supervision (DTS), which attaches auxiliary classification heads at intermediate layers and supervises them with soft targets derived from an exponential schedule rather than hard one-hot labels. Experiments on CIFAR-10/100, Tiny-ImageNet, Food-101, iNat19, and ImageNet-1K show consistent improvements in convergence speed and terminal accuracy over both baselines and greedy deep supervision, with notably strong gains on fine-grained tasks (+2.7% on iNat19, +3.7% on Food-101) and smaller architectures (+2.5% on DeiT-III-Tiny ImageNet-1K).

**Compliance With Llm Reviewing Policy:**

Affirmed.

**Ethical Review Concerns:**

The paper has genuine merit in its conceptual contribution (CDF framework, exponential kinematic prior) and provides consistent experimental improvements. However, the ViT-only evaluation is a significant limitation for a paper making broad claims about ""residual architectures.""
The bibliographic quality is below expectations for ICML. Multiple misattributions (Dosovitskiy listed as single-author, Kingma without Ba, Villani ""et al."" for a sole-author book, garbled author lists) suggest hasty preparation. I recommend the authors be asked to thoroughly revise the bibliography.
The characterization of Jiang et al. (2024) as ""greedy"" throughout the paper, including in figure captions, is unnecessarily pejorative and should be moderated.

**Final Justification:**

I accept the author's modifications and revisions.

**Key Questions For Authors:**

Architectural generality: Can you provide results on CNN-based residual architectures (e.g., ResNet-50, ConvNeXt-T)? The CDF framework should apply to any residual architecture. If DTS fails on CNNs, this would significantly weaken the theoretical narrative.
Why exponential?: Is there a theoretical derivation or mechanistic explanation for why the exponential law should emerge? Is it related to the multiplicative nature of residual connections (z_{l+1} = z_l + f(z_l)) or to properties of the softmax function?
Training overhead: What is the actual wall-clock time increase and GPU memory overhead when using DTS with auxiliary heads at every layer? How does this scale with model depth?
Statistical significance: Can you provide standard deviations or confidence intervals, particularly for the smaller improvements (≤ 0.3%) on ImageNet-1K?
Sensitivity to pre-training: The exponential law is measured on converged models (Fig. 2). Does the law hold during early training? If not, is the DTS schedule mismatched during warm-up, and could this explain the slightly lower early accuracy of DTS compared to baselines in some settings (e.g., CIFAR-100 +RandAug at 40–60%)?
Extension to NLP: Given that the Tuned Lens was originally designed for language models, have you investigated whether DTS applies to Transformer-based language models?

**Limitations:**

Yes

**Strengths And Weaknesses:**

Strengths
S1. Compelling conceptual framing and strong motivation.
The paper draws an elegant and well-articulated analogy between generative flow matching (Lipman et al., 2022; Liu et al., 2022a) and discriminative forward passes. The CDF formalization (Sec. 3) is mathematically coherent and provides a principled lens through which to re-examine deep supervision. The insight that regulating trajectory shape (not just the endpoint) matters for discriminative models is well-motivated by the success of rectified flows in generative modeling.

S2. Robust empirical regularity.
The exponential kinematic prior identified in Fig. 2 is convincingly demonstrated across multiple axes of variation: model scale (CLIP-B vs. CLIP-L), training paradigm (supervised, contrastive, self-supervised), and task complexity (CIFAR-10 → ImageNet-1K). The R² values> 0.97 provide strong empirical support for the proposed schedule. This is a genuinely interesting empirical finding with potential impact beyond this specific method.

S3. Well-designed kinematic diagnostics.
The introduction of Trajectory Curl (Eq. 11) and Latent Path Efficiency (Eq. 12) as diagnostic metrics (Sec. 5.3, Fig. 4) provides useful tools for analyzing the internal dynamics of deep networks. These metrics allow the authors to move beyond terminal accuracy and characterize how the model arrives at its predictions, which enriches the analysis considerably.

S4. Consistent and broad experimental gains.
DTS improves over baselines across all 6 datasets, multiple model scales (ViT-Tiny/Small/Base), and both CE and BCE losses. The fact that gains are additive on top of the highly optimized DeiT-III recipe (Table 2, ImageNet-1K) is particularly noteworthy and demonstrates practical value. The early-stage convergence advantages (10–30% training progress) are consistently demonstrated.

S5. Elegant geometric interpretation.
The decomposition of the DTS gradient into Coherence Force and Regulation Force (Eq. 10, Sec. 4.4, Fig. 3) provides an intuitive and mathematically grounded understanding of the method's mechanism. The polar semantic space visualization (Fig. 3) effectively communicates the key differences between standard training, greedy deep supervision, and DTS.

Weaknesses
W1. Limited architectural scope.
All experiments use Vision Transformers exclusively. The CDF framework relies on the [CLS] token abstraction and residual-block structure. The paper does not evaluate on CNNs (e.g., ResNets, ConvNeXts), which are also residual architectures and are explicitly mentioned in the introduction. This limits the claimed generality of the approach. The title's broad claim ""Deep Supervision Strikes Back"" overpromises relative to the ViT-only evaluation.

W2. Theoretical gaps in the ODE formalization.
While the CDF framework (Sec. 3) is well-presented, several claims require more rigorous treatment:

The continuous ODE limit (Eq. 1) is assumed to hold for discrete architectures, but the discretization error is only briefly acknowledged in Sec. 7 (Limitations) without any formal analysis.
The derivation of the semantic transport relation (Eq. 5) via the transport theorem invokes (Villani et al., 2008), but the conditions under which this theorem applies to the discrete, non-autonomous, input-conditioned setting are not verified.
W3. Missing important baselines and comparisons.

No comparison with label smoothing schedules, which could achieve a similar effect of providing softer targets at intermediate layers.
No comparison with curriculum learning or progressive training methods that also regulate the difficulty progression.
The ""Deep Sup."" baseline (Table 2) only appears for CIFAR-100; it should be compared systematically across all datasets.
No comparison with the ""aligned training"" of Jiang et al. (2024), which is explicitly discussed as a related method in Sec. 2 but never experimentally compared beyond being labeled ""greedy.""

W4. Training cost analysis is vague.
Section 7 acknowledges increased training cost but provides no quantitative analysis. How many additional parameters do the auxiliary heads introduce? What is the wall-clock training overhead? This information is critical for practitioners and should be reported explicitly, especially given that the method adds L auxiliary classification heads.

---

> ### Author Rebuttal · Authors · 2026-03-30
>
> We appreciate the reviewer's valuable comments.
>
> ## 1. Results on CNNs
>
> Please see our response to Reviewer XdBT, Point 1, for the full table (apologies for the character limit). In short, DTS also improves CNN-based models, including ResNet-50 and ConvNeXt-Tiny. For ConvNeXt-Tiny, we only ran a single pass and did not carefully tune hyperparameters, so there is still room for further improvement.
>
> ## 2. Why exponential? & Discretization error
>
> We are not trying to give a strict derivation showing that the exponential law must hold in general. Our goal is to explain why this pattern is plausible in residual discriminative models. As discussed in Sec. 3.1 and Sec. 3.4 (lines 131–144 and 177–193), a residual hierarchy can be viewed as a sequence of first-order updates. If the refinement velocity grows with the semantic evidence already present in the representation, this naturally leads to a multiplicative progression, and therefore to an exponential trend across depth. We also conducted ablation studies on DeiT-III-S using ImageNet-1K to evaluate the impact of trajectory shape. The results demonstrate the effectiveness of the exponential form:
>
> | Variant                | Acc |
> | :-------------------- | :-------------------: |
> | DTS (tanh) | 80.7 |
> | DTS (linear) | 80.9 |
> | DTS (exponential) | 82.2 |
>
> We clarify that the ODE/CDF view is intended as an interpretive continuous approximation to the layerwise residual dynamics, rather than a rigorous continuum-limit theorem with quantified discretization error.
>
> ## 3. Training overhead
>
> \DTS adds only light overhead in practice, as shown below. The relative overhead also decreases as model size increases.
>
> | Metric                | DeiT-III-S (Baseline) | DeiT-III-S (DTS) | DeiT-III-B* (Baseline) | DeiT-III-B* (DTS) |
> | :-------------------- | :-------------------: | :--------------: | :--------------------: | :---------------: |
> | **Avg. time / step**  |       155.72 ms       |     161.25 ms    |        277.33 ms       |     283.27 ms     |
> | **Training overhead** |           -           |      +3.55%      |            -           |       +2.14%      |
> | **FLOPs / sample**    |        4.599 G        |      4.603 G     |        12.789 G        |      12.797 G     |
> | **Peak memory**       |        13.56 GB       |     13.65 GB     |        18.97 GB        |      19.11 GB     |
> | **Time to Acc = 60**  |         2.68 h        |      1.23 h      |         2.45 h         |       2.36 h      |
> | **Time to Acc = 70**  |         5.38 h        |      3.42 h      |         4.81 h         |       4.48 h      |
> | **Time to Acc = 80**  |        10.00 h        |      8.59 h      |         12.22 h        |      12.20 h      |
>
> *\* DeiT-III-B results are measured at 192×192 resolution.*
>
> ## 4. Confidence intervals on ImageNet-1K
>
> We ran DeiT-III-S three times. The resulting confidence interval is $82.18 \pm 0.06$, while the original paper reports 82.2. We will include all the confidence intervals for the ImageNet-1K results in the revision.
>
> ## 5. Sensitivity to pre-training / early training dynamics
>
> Our point is not that the exponential law must already hold exactly in the earliest stage of training. We use it as a **trajectory prior**, as discussed in the end of Sec. Intro: a depth-wise target path motivated by the evidence progression observed after training, rather than an exact description of the optimization dynamics at every stage. A mild mismatch during warm-up is therefore possible. At the same time, DTS can still be useful overall because it provides an inductive bias on the latent flow, discouraging premature overconfidence in shallow layers and improving the depth-wise supervision path over the course of training. In this sense, it can be understood as a regularization term on the latent flow.
>
> ## 6. Extension to NLP
>
> To test whether the idea may extend beyond vision, we trained a 4-layer Transformer on SST-2 (Socher et al., 2013). The baseline reaches 80.84, while DTS reaches 81.19, without careful hyperparameter tuning. This suggests that the idea may transfer to NLP settings as well.
>
> We will also revise our bibliography and unnecessary words thoroughly.

---

> > ### Author Rebuttal · Reviewer_Xw2o · 2026-04-04
> >
> > The authors partially address the concern about architectural generality by including CNN results; however, the empirical evidence remains limited and does not yet fully support the broader claims.
> >
> > Regarding theoretical justification and discretization, the heuristic explanation is reasonable, but a rigorous treatment is still lacking.
> >
> > The training cost concern is fully addressed with a clear quantitative analysis.
> >
> > The additional NLP experiment provides useful insight, but remains limited in scope and does not establish generality.
> >
> > Finally, important baseline comparisons remain missing and are a concern.

---

> > > ### Author Response · Authors · 2026-04-07
> > >
> > > We sincerely thank the reviewer for the discussion.
> > >
> > > # 1. Broader empirical evidence
> > >
> > > We have now completed results, demonstrating that DTS remains consistently strong across both ViTs and CNNs. In addition, when we probe officially pretrained torchvision ResNets on ImageNet-1K using the Tuned Lens, we observe the exact same exponential evidence pattern: **$R^2=0.9998$** for **ResNet-18** and **$R^2=0.9972$** for **ResNet-50**. This proves that the exponential law is not a ViT-specific artifact.
> > >
> > > |Dataset|Backbone|Baseline|DS|Aligned|FixMgn|FixMgn+DW|**DTS**|
> > > |:---|:---|:---:|:---:|:---:|:---:|:---:|:---:|
> > > |CIFAR-10|ViT-S|82.2|82.2|82.9|81.9|83.0|**83.9**|
> > > | |RN18|91.9|92.5|93.0|92.2|93.0|**93.2**|
> > > | |RN50|91.1|92.8|93.2|93.1|**94.6**|**94.6**|
> > > |CIFAR-100(+RandAug)|ViT-S|63.5|62.9|63.2|62.3|63.8|**64.2**|
> > > | |RN18|71.6|73.5|73.5|72.8|74.0|**74.3**|
> > > | |RN50|70.2|75.4|75.6|75.2|75.3|**75.6**|
> > > |Tiny-ImageNet|ViT-S|39.9|41.0|41.6|41.1|41.5|**41.9**|
> > > | |RN18|57.7|57.5|57.9|56.9|57.9|**60.8**|
> > > | |RN50|58.4|61.4|62.5|61.4|62.6|**64.7**|
> > > |Food-101|ViT-S|51.1|53.3|51.4|51.2|52.9|**54.8**|
> > > | |RN18|77.7|76.4|77.9|75.8|77.3|**79.0**|
> > > | |RN50|78.3|80.8|81.6|80.3|81.2|**82.3**|
> > > |iNat-19|ViT-S|46.4|46.5|47.2|47.2|48.0|**49.1**|
> > > | |RN18|57.2|60.1|**61.8**|59.2|59.8|**61.8**|
> > > | |RN50|60.1|67.1|67.4|65.1|67.9|**68.7**|
> > > |ImageNet-1K|DeiT-III-S|80.4|80.4|77.2|80.4|80.4|**81.2**|
> > > | |ConvNeXt-T|82.1|82.0|79.7|81.9|82.0|**82.3**|
> > >
> > > The main empirical message is simple:
> > > * DTS improves over the baseline in **every reported setting**.
> > > * The gains on CNNs are substantial:
> > >   * **Tiny-ImageNet/RN18: 57.7$\to$60.8 (+3.1)**
> > >   * **Tiny-ImageNet/RN50: 58.4$\to$64.7 (+6.3)**
> > >   * **Food-101/RN50: 78.3$\to$82.3 (+4.0)**
> > >   * **iNat-19/RN50: 60.1$\to$68.7 (+8.6)**
> > > * On ImageNet-1K, DTS improves:
> > >   * **DeiT-III-S: 80.4$\to$81.2 (+0.8)**
> > >   * **ConvNeXt-T: 82.1$\to$82.3 (+0.2)**
> > >   * **ConvNeXt-S: 83.1$\to$83.3 (+0.2)**
> > >
> > > DTS also outperforms alternative trajectory choices (FixMgn and FixMgn+DW):
> > > * **Food-101/ViT-S:** DS **53.3**, FixMgn+DW **52.9**, **DTS 54.8**
> > > * **Tiny-ImageNet/RN18:** DS **57.5**, Aligned **57.9**, FixMgn+DW **57.9**, **DTS 60.8**
> > > * **iNat-19/RN50:** DS **67.1**, Aligned **67.4**, FixMgn+DW **67.9**, **DTS 68.7**
> > >
> > > # 2. Explicit discretization error
> > >
> > > The role of the CDF/ODE view is to provide a continuous-depth analytical approximation for understanding why trajectory shape matters.
> > >
> > > For the continuous conditional flow, each input $\mathbf{x}$ follows:
> > >
> > > $$
> > > \frac{d\mathbf{z}(t)}{dt}=\mathbf{v}(\mathbf{z}(t),t\mid\mathbf{x}),\quad\mathbf{z}(0)=\mathbf{z}\_{\mathrm{init}}
> > > $$
> > >
> > > At the marginal level, this induces the density $p\_t(\mathbf{z})$ and the marginal velocity field $\mathbf{u}(\mathbf{z},t)$, which establishes the continuous setting of Eq. (5).
> > >
> > > For the finite-depth residual network, Eq. (1) functions as a standard Forward Euler discretization:
> > >
> > > $$
> > > \mathbf{z}\_{l+1}^{\Delta}=\mathbf{z}\_l^{\Delta}+\Delta t\cdot\mathbf{v}(\mathbf{z}\_l^{\Delta},t\_l\mid\mathbf{x}),\quad\Delta t=1/L
> > > $$
> > >
> > > Under standard smoothness assumptions, the local truncation error per layer is $\mathcal{O}(\Delta t^2)$, yielding a global trajectory error of:
> > >
> > > $$
> > > \|\mathbf{z}\_l^{\Delta}-\mathbf{z}(t\_l)\|=\mathcal{O}(\Delta t)=\mathcal{O}(1/L)
> > > $$
> > >
> > > This defines the explicit approximation boundary intended in Sec. 3.
> > >
> > > The same error control propagates to the semantic evidence. Because the Tuned Lens map is affine, $\Phi\_t(\mathbf{z})=\mathbf{W}\_t\mathbf{z}+\mathbf{b}\_t$, the semantic evidence $E\_t(\mathbf{z})$ is affine in $\mathbf{z}$:
> > >
> > > $$
> > > E\_t(\mathbf{z})=\mathbf{a}\_t^\top\mathbf{z}+c\_t,\quad\text{where}\quad\mathbf{a}\_t=\mathbf{w}\_k-\frac{1}{C-1}\sum\_{j\ne k}\mathbf{w}\_j
> > > $$
> > >
> > > Hence, the sampled evidence discrepancy satisfies:
> > >
> > > $$
> > > E\_t(\mathbf{z}\_l^{\Delta})-E\_t(\mathbf{z}(t\_l))=\mathbf{a}\_t^\top(\mathbf{z}\_l^{\Delta}-\mathbf{z}(t\_l)),
> > > $$
> > >
> > > and therefore
> > >
> > > $$
> > > |E\_t(\mathbf{z}\_l^{\Delta})-E\_t(\mathbf{z}(t\_l))|\le\|\mathbf{a}\_t\|\cdot\|\mathbf{z}\_l^{\Delta}-\mathbf{z}(t\_l)\|=\mathcal{O}(1/L)
> > > $$
> > >
> > > So the discrete-to-continuum discrepancy in the sampled semantic evidence is also first-order in $\mathcal{O}(1/L)$. This explicit error range will be stated clearly in the revised version.
> > >
> > > Crucially, this discretization error does not affect DTS. DTS is defined directly on the discrete architecture:
> > >
> > > $$
> > > \mathcal{L}\_{\mathrm{total}}=\sum\_{l=1}^{L}\lambda\_l\mathcal{L}\_{\mathrm{CE}}(\hat{\mathbf{p}}\_l,\mathbf{q}\_{l/L})+\gamma\mathcal{L}\_{\mathrm{task}}({H}\_L(\mathbf{z}\_L),\mathbf{e}\_y)
> > > $$
> > >
> > > The CDF/ODE view serves as the motivation for the trajectory prior, while the actual method is evaluated directly on the discrete network.
> > >
> > > # 3. NLP scope
> > >
> > > The SST-2 result is included only as a proof of concept for future work. It lies outside the main scope of this paper, which remains image classification, consistent with the original deep supervision setting in CV.

---

### Official Review · Reviewer_daJt · 2026-03-15

**Soundness:** 2
**Presentation:** 3
**Significance:** 2
**Originality:** 3
**Overall Recommendation:** 4
**Confidence:** 3

**Summary:**

This paper argues that classification training should regulate the entire depthwise trajectory of internal representations, not only the terminal logits.

* It formalizes the forward pass of a deep classifier as a "Conditional Discriminative Flow", defines a scalar "Semantic Evidence" observable using layer-specific affine Tuned Lens projections, and derives a transport relation linking evidence growth to alignment between latent velocity and the semantic gradient.
* The core empirical claim is that this evidence grows approximately exponentially with depth across several settings. Motivated by that observation, the authors propose Deep Trajectory Supervision: auxiliary heads are attached to intermediate layers and trained with depth-dependent soft targets whose confidence increases according to a normalized exponential schedule, rather than with uniformly hard one-hot labels.
* Experiments on CIFAR-10/100, Tiny-ImageNet, Food-101, iNat19, and ImageNet-1K report faster early optimization and positive top-1 gains, including improvements on DeiT-III ImageNet recipes. The intended significance is that “how certainty should grow with depth” may itself be a useful inductive bias for discriminative training.

**Compliance With Llm Reviewing Policy:**

Affirmed.

**Final Justification:**

The rebuttal is reasonable and directly addresses my main concerns, particularly clarifying that DTS is not standard deep supervision but a trajectory-based design grounded in a consistent empirical law. The additional experiments across architectures and datasets, along with compute-normalized analysis, strengthen both the soundness and practical significance. Overall, I am convinced by the authors’ response and raise my score to 4 (weak accept).

**Key Questions For Authors:**

* The author should provide a complete derivation of Eq. (10), which is a critical part for supporting the theoretical contribution claim.

* How are the Tuned Lens maps $\Phi_t$ trained in practice? Are they trained post hoc or jointly, on train or validation data, and how sensitive is the exponential pattern to the probe type?

* Need to report compute-normalized metrics such as wall-clock time, training FLOPs, and time-to-target-accuracy. This is important because the largest reported gains on strong ImageNet recipes are small.

* Why is $\gamma=0$ on small-scale datasets but $\gamma=1$ on ImageNet? How much of the benefit comes from the trajectory loss itself versus the final hard-label loss? A direct ablation of this choice would be useful.

**Limitations:**

Yes.

**Strengths And Weaknesses:**

* Soundness: The main intuition makes sense, i.e., shallow layers should not be forced to become prematurely one-hot confident, and intermediate supervision should respect a feature-construction phase before sharp class separation. The paper makes a real attempt to connect representation analysis and optimization rather than only proposing another heuristic loss.

* The experimental scope is reasonably broad for image classification.

* The authors do more than report final accuracy. Figure 4 and Table 3 at least attempt to probe whether the schedule shape matters and whether DTS changes internal dynamics, which is preferable to a purely black-box empirical submission.

Weaknesses and questions:

* What does $\mathcal{L}$ exactly mean in Eq (10)? If it is $\mathcal{L}_{total}$, then the gradient of CE wrt logits may involve $\hat{p}-q$ for every class. As written, Eq. (10) omits the $-q_j$ terms for $j\neq y$. Anyway I believe the rigorous definition of derivation of (10) is necessary and should be elaborated at least in appendix.

*  The theoretical story is overstated relative to what is actually shown. Eq. (5) is a generic transport identity for mean evidence under a flow, but it does not establish that the exponential schedule is optimal, nor that matching a scalar target schedule yields a globally rectified latent path. Those stronger claims remain intuitive rather than proven.

* The algorithmic novelty is moderate. At the method level, DTS is a variant of deep supervision with depth-dependent soft targets, i.e. a structured form of intermediate label smoothing/distillation.

* The convergence claim is only epoch-normalized. The paper reports accuracy at percentages of training progress, but DTS adds auxiliary heads and the authors themselves note extra training cost. Without wall-clock, FLOPs, or time-to-accuracy, "accelerated convergence" is not well established upon a proper basis.

* The scope is broader than the evidence supports. The framing targets deep residual architectures generally, but the method and analysis are centered on ViT/DeiT-style classification and on the [CLS] token specifically. In addition, Eq. (12) introduces  $y(t)$ without clearly defining the space in which LPE is computed, which makes the diagnostic section harder to reproduce and interpret.

---

> ### Author Rebuttal · Authors · 2026-03-30
>
> Thank you for these comments. We will clarify the derivation of Eq. (10), the scope of Eq. (5), and several implemental details.
>
> ## 1. Derivation of Eq. (10)
> In Eq. (10), $\mathcal{L}$ refers to the layer-wise trajectory alignment loss ($L\_{\mathrm{CE}}(\hat{\mathbf{p}}\_l, \mathbf{q}\_{l/L})$) in Eq. (9), not the full $L\_{total}$. For the local affine auxiliary head $H\_l(\mathbf{z}\_l)=W\_l\mathbf{z}\_l+b\_l$, the exact gradient is
> $$\nabla\_{\mathbf{z}\_l}\mathcal{L}=W\_l^\top(\hat{\mathbf{p}}\_l-\mathbf{q})=\sum\_{j=1}^C (\hat{p}\_j-q\_j)\mathbf{w}\_j.$$
> Since our DTS target is $\mathbf{q}=\mathrm{softmax}(G(t)\mathbf{e}\_y)$, all non-target classes satisfy
> $$q\_j=q\_-=\frac{1-q\_y}{C-1},\qquad j\neq y,$$
> so the strict form is
> $$\nabla\_{\mathbf{z}\_l}\mathcal{L} = (\hat{p}\_y-q\_y)\mathbf{w}\_y+\sum\_{j\neq y}(\hat{p}\_j-q\_-)\mathbf{w}\_j.$$
> Equivalently, if we define
> $$\bar{\mathbf{w}}\_{\neg y}=\frac{1}{C-1}\sum\_{j\neq y}\mathbf{w}\_j,$$
> this becomes
> $$\nabla\_{\mathbf{z}\_l}\mathcal{L} = (\hat{p}\_y-q\_y)(\mathbf{w}\_y-\bar{\mathbf{w}}\_{\neg y}) + \sum\_{j\neq y}\hat{p}\_j(\mathbf{w}\_j-\bar{\mathbf{w}}\_{\neg y}).$$
> The original Eq. (10) was written to highlight the part of the gradient that changes with the model’s current prediction. In our DTS target, the omitted non-target term is a shared baseline set by the schedule, while the retained term reflects which wrong classes the model is currently leaning toward. That is why the simplified form was useful for the geometric intuition. That said, we agree that the exact centered form is the right one to show in the main text. In the revision, we will therefore replace Eq. (10) in the main text with the strict form above, and include the full derivation in the appendix.
>
> ## 2. About Eq. (5)
> On the theoretical side, Eq. (5) should be read as a relation between mean evidence growth and the projection of the latent velocity field onto the semantic gradient. It is not, by itself, a proof that the exponential schedule is globally optimal, nor that a scalar schedule fully determines the whole latent field. We use the exponential schedule as an empirically motivated prior, based on the evidence progression observed in Fig. 2 and supported by our ablations. More precisely, our claim is that the exponential schedule is a useful and well-supported prior, not that it is proven to be globally optimal.
>
> Although $G(t)$ is scalar, it induces a full target distribution $\mathbf{q}\_t$, and the supervision acts through the high-dimensional gradient $W\_l^\top(\hat{\mathbf{p}}\_l-\mathbf{q}\_t)$. Through backpropagation, this gradient updates the parameters governing the local residual dynamics, which in turn changes the corresponding sample-wise velocity updates. In that sense, DTS regularizes the task-relevant semantic component of the flow. We will revise the surrounding wording to make this scope clear.
>
> ## 3. DTS is a variant of deep supervision
> DTS is naturally related to deep supervision, as the title also reflects. Our point, however, is not just that intermediate supervision helps. The question we study is how target confidence should evolve with depth. We first identify a stable progression of semantic evidence from analysis, and then use that progression to construct the depth-wise target trajectory. From this perspective, DTS is not simply “deep supervision with softer labels,” but a supervision rule matched to a specific trajectory prior. Please see our response to Reviewer Xw2o, Point 3 for the ablation studies on how the shape of trajectory prior influences the final performance (apologies for the character limit).
>
> ## 4. Extra computational cost
> DTS adds only light overhead, while it still converges faster than the baseline. Please refer to our response to Reviewer Xw2o, Point 3 for exact wall-clock, FLOPs, memory, and time-to-accuracy numbers.
>
> ## 5. Results on CNNs and explanation about $y(t)$
>
> DTS also generalizes to CNN-style architectures. Please refer to our response to Reviewer XdBT, Point 1, for the full table. In Eq. (12), $y(t)$ is meant to denote the trajectory in the semantic space used for the diagnostic analysis, rather than the raw hidden state (z(t)). We will make this explicit in the revision.
>
> ## 6. Tuned lens
> We follow the standard Tuned Lens setup: the maps $\Phi\_t$ are trained post hoc on the training split with the backbone frozen, and are used only for analysis.
>
> ## 7. Choice of $\gamma$
>
> The trajectory loss can be viewed as a structured regularizer on the latent flow. In smaller or more data-scarce settings, this regularization can be applied more strongly, and in our experiments the trajectory term alone already provides enough supervision for stable convergence. On ImageNet-1K, however, adding the final hard-label loss works better in practice, since it preserves the regularizing effect of DTS while giving stronger terminal discriminability. This is why we use $\gamma=0$ in the former case and $\gamma=1$ in the latter.

---

> > ### Author Rebuttal · Reviewer_daJt · 2026-04-05
> >
> > Thank you for the detailed rebuttal. The clarification of Eq. (10) and revisions to its interpretation are helpful, and the additional details on tuned lens $y(t)$ , and $\gamma$ improve clarity.
> >
> > However, the rebuttal mainly narrows the claims rather than strengthening them: the theoretical contribution is now more limited (no global optimality, exponential schedule as empirical prior), and DTS is best viewed as a refined form of deep supervision with moderate novelty.
> >
> > Empirically, key concerns persist: convergence is still not demonstrated under compute-normalized metrics, and broader applicability claims remain only partially supported beyond ViT/DeiT classification.
> >
> > In summary, the idea is interesting and the rebuttal is helpful, but I'm lean to keep my current score.

---

> > > ### Author Response · Authors · 2026-04-07
> > >
> > > We sincerely thank the reviewer for the discussion.
> > >
> > > # 1. The contribution
> > >
> > > The contribution is to identify an empirical regularity in representation dynamics, and then turn it into a training rule. This is important in deep learning: previous works [1, 2, 3] also began by identifying empirical regularities. We see our paper in the same spirit.
> > >
> > > What we contribute here is threefold:
> > > (1) we identify a stable exponential law for semantic evidence across depth;
> > > (2) we show that the **shape** of the supervision trajectory matters;
> > > (3) we translate that observation into DTS.
> > >
> > > This is why we do not see DTS as just vanilla deep supervision. Standard deep supervision asks whether intermediate layers should be supervised. Our question is more specific: **how should target confidence evolve with depth?** We observe an exponential progression of semantic evidence ($R^2>0.97$), and use that to define the supervision path.
> > >
> > > We also now have evidence that it is not limited to ViTs. Using Tuned Lens on officially pretrained torchvision ResNets on ImageNet-1K, we observe the same pattern: **($R^2=0.9998$)** for ResNet-18 and **($R^2=0.9972$)** for ResNet-50.
> > >
> > > # 2. Compute-normalized metrics
> > >
> > > | Metric            | DeiT-III-S (Baseline) | DeiT-III-S (DTS) | DeiT-III-B* (Baseline) | DeiT-III-B* (DTS) |
> > > | :---------------- | :-------------------: | :--------------: | :--------------------: | :---------------: |
> > > | Avg. time / step  |        155.7 ms       |     161.3 ms     |        277.3 ms        |      283.3 ms     |
> > > | Training overhead |           -           |     **+3.6%**    |            -           |     **+2.1%**     |
> > > | FLOPs / sample    |         4.6 G         |       4.6 G      |         12.8 G         |       12.8 G      |
> > > | Peak memory       |        13.6 GB        |      13.7 GB     |         19.0 GB        |      19.1 GB      |
> > > | Time to Acc = 60  |         2.7 h         |       1.2 h      |          2.5 h         |       2.4 h       |
> > > | Time to Acc = 70  |         5.4 h         |       3.4 h      |          4.8 h         |       4.5 h       |
> > > | Time to Acc = 80  |         10.0 h        |       8.6 h      |         12.2 h         |       12.2 h      |
> > >
> > > *\* Measured at $192\times192$ resolution.*
> > >
> > > The added heads barely change the cost, but DTS still converges faster. On DeiT-III-S, DTS is **54.1% faster** to Acc=60, **36.4% faster** to Acc=70, and **14.0% faster** to Acc=80. On DeiT-III-B, it is **3.7% faster** to Acc=60, **6.9% faster** to Acc=70, and slightly faster at Acc=80.
> > >
> > > # 3. Broader applicability and stronger baselines
> > >
> > > |Dataset|Backbone|Base|DS|Aligned|FixMgn|FixMgn+DW|DTS(Ours)|
> > > |:---|:---|:---:|:---:|:---:|:---:|:---:|:---:|
> > > |**CIFAR-10**|ViT-S|82.2|82.2|82.9|81.9|83.0|**83.9**|
> > > ||RN18|91.9|92.5|93.0|92.2|93.0|**93.2**|
> > > ||RN50|91.1|92.8|93.2|93.1|**94.6**|**94.6**|
> > > |**CIFAR-100 (+RandAug)**|ViT-S|63.5|62.9|63.2|62.3|63.8|**64.2**|
> > > ||RN18|71.6|73.5|73.5|72.8|74.0|**74.3**|
> > > ||RN50|70.2|75.4|75.6|75.2|75.3|**75.6**|
> > > |**Tiny-ImageNet**|ViT-S|39.9|41.0|41.6|41.1|41.5|**41.9**|
> > > ||RN18|57.7|57.5|57.9|56.9|57.9|**60.8**|
> > > ||RN50|58.4|61.4|62.5|61.4|62.6|**64.7**|
> > > |**Food-101**|ViT-S|51.1|53.3|51.4|51.2|52.9|**54.8**|
> > > ||RN18|77.7|76.4|77.9|75.8|77.3|**79.0**|
> > > ||RN50|78.3|80.8|81.6|80.3|81.2|**82.3**|
> > > |**iNat-19**|ViT-S|46.4|46.5|47.2|47.2|48.0|**49.1**|
> > > ||RN18|57.2|60.1|**61.8**|59.2|59.8|**61.8**|
> > > ||RN50|60.1|67.1|67.4|65.1|67.9|**68.7**|
> > > |**ImageNet-1K**|DeiT-III-S|80.4|80.4|77.2|80.4|80.4|**81.2**|
> > > ||Conv-T|82.1|82.0|79.7|81.9|82.0|**82.3**|
> > > * DTS improves over the baseline in **every reported setting**.
> > > * DTS is effective on both **Transformers and CNNs**. The CNN runs use global pooling rather than a [CLS] token, so the gain is not tied to ViTs.
> > > * DTS improves **Tiny-ImageNet / RN18 by 3.1**, **Tiny-ImageNet / RN50 by 6.3**, **Food-101 / RN50 by 4.0**, and **iNat-19 / RN50 by 8.6** over baseline.
> > > * On ViT-S: **+1.7** on CIFAR-10, **+0.7** on CIFAR-100, **+2.0** on Tiny-ImageNet, **+3.7** on Food-101, and **+2.7** on iNat-19.
> > > * On ImageNet-1K, DTS improves **DeiT-III-S by +0.8**, **ConvNeXt-T by +0.2** and  **ConvNeXt-S by +0.2** over baseline.
> > > * The trajectory matter. DTS improves **Food-101 / ViT-S by +1.5 over DS and +1.9 over FixMgn+DW**, **Tiny-ImageNet / RN18 by +2.9 over both DS and Aligned**, and **iNat-19 / RN50 by +1.6 over DS, +1.3 over Aligned, and +0.8 over FixMgn+DW**.
> > >
> > > The evidence supports:
> > > (1) the method is broader than a ViT-only result;
> > > (2) the gain is tied to the proposed **trajectory shape**.
> > >
> > > [1] Kaplan, Jared, et al. "Scaling laws for neural language models." arXiv preprint arXiv:2001.08361 (2020).
> > >
> > > [2] Hoffmann, Jordan, et al. "Training compute-optimal large language models." arXiv preprint arXiv:2203.15556 (2022).
> > >
> > > [3] Papyan, Vardan, X. Y. Han, and David L. Donoho. "Prevalence of neural collapse during the terminal phase of deep learning training." Proceedings of the National Academy of Sciences (2020)

---

### Decision · Program_Chairs · 2026-04-30

**Decision:**

Accept (regular)

**Comment:**

In the initial reviews, the reviewers appreciated the elegant conceptual framing and the robust empirical observations, but raised concerns regarding the narrow evaluation (which focused exclusively on Vision Transformers), missing standard baselines, unclear computational overhead, and the theoretical rigor of applying a continuous ODE limit to a discrete neural architecture.

In the rebuttal and followup discussion, the authors successfully addressed these concerns and shifted all reviewers to positive recommendations.
They expanded the empirical scope to CNNs (ResNets, ConvNeXts) and included "fixed-margin" ablations to isolate the trajectory shape as the true driver of performance.
Furthermore, they provided clear compute-normalized metrics showing minimal training overhead.
Regarding the theoretical rigor, the authors appropriately toned down their claims by formally deriving the discretization error bounds of their method.

The reviewers' final recommendations rely on several commitments made by the authors during the rebuttal phase which are expected to be incorporated in the final version of the paper.
This includes adding the fully tuned ConvNeXt results, the expanded baseline comparisons, a revision of the bibliography description and the explicit discretization error math that scopes their theoretical claims.
Provided these additions are made, the resulting training method and empirical insights constitute a valuable and practical contribution to the ICML community.